# Molecular basis for inner kinetochore configuration through RWD domain–peptide interactions

Florian Schmitzberger[1,2,*,†] iD, Magdalena M Richter[3,‡] iD, Yuliya Gordiyenko[4,§], Carol V Robinson[4] iD, Michał Dadlez[3,5] & Stefan Westermann[2,†] iD

## Abstract

Kinetochores are dynamic cellular structures that connect chromosomes to microtubules. They form from multi-protein assemblies that are evolutionarily conserved between yeasts and humans. One of these assemblies—COMA—consists of subunits Ame1[CENP-U], Ctf19[CENP-P], Mcm21[CENP-O] and Okp1[CENP-Q]. A description of COMA molecular organization has so far been missing. We defined the subunit topology of COMA, bound with inner kinetochore proteins Nkp1 and Nkp2, from the yeast *Kluyveromyces lactis*, with nano-flow electrospray ionization mass spectrometry, and mapped intermolecular contacts with hydrogen-deuterium exchange coupled to mass spectrometry. Our data suggest that the essential Okp1 subunit is a multi-segmented nexus with distinct binding sites for Ame1, Nkp1-Nkp2 and Ctf19-Mcm21. Our crystal structure of the Ctf19-Mcm21 RWD domains bound with Okp1 shows the molecular contacts of this important inner kinetochore joint. The Ctf19-Mcm21 binding motif in Okp1 configures a branch of mitotic inner kinetochores, by tethering Ctf19-Mcm21 and Chl4[CENP-N]-Iml3[CENP-L]. Absence of this motif results in dependence on the mitotic checkpoint for viability.

**Keywords** CCAN; COMA; kinetochore; RWD domain; structural biology
**Subject Categories** Cell Cycle; Structural Biology
**The EMBO Journal (2017) 36: 3458–3482**

## Introduction

Kinetochores are the specialized, dynamic macromolecular structures that connect chromatin to spindle microtubules, mediating chromosome segregation and sister chromatid segregation during meiosis and mitosis in eukaryotes. In *Saccharomyces cerevisiae* and other budding yeasts, kinetochores assemble on ~120–300 base pairs of specific DNA—the centromere, and connect to a single microtubule (Winey *et al*, 1995). Kinetochores from other fungi, many animals and plants presumably consist of modular repeats of the kinetochore entity found in budding yeasts (Zinkowski *et al*, 1991; Joglekar *et al*, 2008; Schleiffer *et al*, 2012; Westermann & Schleiffer, 2013; Weir *et al*, 2016).

The ~45 unique proteins that compose the core *S. cerevisiae* kinetochore localize within a distance of ~80 nm between centromere and microtubule (Joglekar *et al*, 2009). Centromere-proximal proteins or microtubule-proximal proteins form inner kinetochore or outer kinetochore, respectively. Many of them organize in separable protein assemblies (De Wulf *et al*, 2003), several of which are present in multiple copies (Joglekar *et al*, 2006; Lawrimore *et al*, 2011). One important inner kinetochore assembly that is conserved between budding yeasts and humans (Schleiffer *et al*, 2012) is COMA (De Wulf *et al*, 2003). COMA's counterpart in mammals is the CENP-O/P/Q/U assembly. COMA connects centromere-associated proteins and outer kinetochore (Hornung *et al*, 2014; Dimitrova *et al*, 2016). COMA includes the proteins "associated with microtubules and essential 1" (Ame1) (Cheeseman *et al*, 2002), "chromosome transmission fidelity 19" (Ctf19) (Hyland *et al*, 1999), "minichromosome maintenance 21" (Mcm21) (Poddar *et al*, 1999) and "outer kinetochore protein 1" (Okp1) (Ortiz *et al*, 1999). These proteins are part of a supramolecular assembly from the inner kinetochore, termed CTF19, which contains eight other subunits that include "non-essential kinetochore protein 1" (Nkp1) and "non-essential kinetochore protein 2" (Nkp2) (Cheeseman *et al*, 2002; De Wulf *et al*, 2003; Schleiffer *et al*, 2012). Most CTF19 subunits are homologous to specific inner kinetochore subunits in mammals (Schleiffer *et al*, 2012). These inner kinetochore subunits form the constitutive centromere-associated network (CCAN) (Foltz *et al*, 2006; Okada *et al*, 2006). CCAN is the structural platform for outer kinetochore assembly (Hori *et al*, 2013; Basilico *et al*, 2014; Weir *et al*, 2016). Because of their overall homology, we use the term "CCAN" to refer to both the CCAN assembly and the CTF19 assembly. The specific functional relevance of several CCAN subunits is

1 Department of Biological Chemistry and Molecular Pharmacology, Harvard Medical School, Boston, MA, USA
2 Research Institute of Molecular Pathology (IMP), Vienna, Austria
3 Institute of Biochemistry and Biophysics, Polish Academy of Sciences, Warsaw, Poland
4 Department of Chemistry, Physical and Theoretical Chemistry Laboratory, University of Oxford, Oxford, UK
5 Institute of Genetics and Biotechnology, Biology Department, Warsaw University, Warsaw, Poland
†Present address: Department of Molecular Genetics 1, University of Duisburg-Essen, Essen, Germany
‡Present address: Department of Genetics, University of Cambridge, Cambridge, UK
§Present address: MRC Laboratory of Molecular Biology, Cambridge, UK
*Corresponding author. Tel: +49 151 67449304; E-mail: florian.schmitzberger@uni-due.de

unclear. Ame1[CENP-U] and Okp1[CENP-Q] (superscripts are human orthologue names; or in the following, in the case of human protein names, budding yeast-orthologue names) are essential for viability (essential) of *S. cerevisiae* (Ortiz *et al*, 1999; Cheeseman *et al*, 2002; Hornung *et al*, 2014).

To understand construction principles, mechanistic functions and assembly of kinetochores, we need to know their subunit structure, subunit contacts and relevance of specific molecular interfaces. We previously determined the structure of the double RWD (D-RWD) domains of Ctf19[CENP-P]-Mcm21[CENP-O] (Schmitzberger & Harrison, 2012). RWD domains are important recurring scaffolds in kinetochores (Wei *et al*, 2006; Ciferri *et al*, 2008; Corbett & Harrison, 2012; Malvezzi *et al*, 2013; Nishino *et al*, 2013; Petrovic *et al*, 2014). How the Ctf19-Mcm21 D-RWD domains are contributing to kinetochore assembly has remained unknown. Molecular subunit arrangement in COMA or CENP-O/P/Q/U, and their role in kinetochore configuration have been poorly understood. In recently reported biochemical reconstitutions of a human kinetochore (Weir *et al*, 2016) or a human CCAN (McKinley *et al*, 2015), CENP-O/P/Q/U was not included.

We found that COMA binds inner kinetochore proteins Nkp1 and Nkp2. To obtain insight into arrangement of and contacts in COMA-Nkp1-Nkp2, we combined data from nanoflow electrospray ionization (nanoflow) mass spectrometry and hydrogen-deuterium exchange coupled to mass spectrometry (deuterium exchange). Through biochemical reconstitution of variant assemblies with truncations of the essential subunits Ame1 and Okp1, we defined inner kinetochore assembly requirements. Data from these experiments were instrumental for our crystal structure determination of Ctf19-Mcm21 with its interacting Okp1 segment. We show that the Ctf19-Mcm21 binding motif in Okp1 is an important tether in the molecular architecture of mitotic inner kinetochores, which is required for kinetochore function.

# Results

## Identification of a COMA-Nkp1-Nkp2 assembly and its molecular composition

We had previously described reconstitution of recombinant full-length COMA from *K. lactis*, by co-expression of *Ame1*, *Ctf19*, *Mcm21* and *Okp1* (Schmitzberger & Harrison, 2012). To understand COMA's integration in the kinetochore, we sought to identify COMA interactions with other kinetochore proteins. We found that two recombinantly produced *K. lactis* CCAN subunits, Nkp1 and Nkp2, form a heterodimer—Nkp1-Nkp2 (Appendix Fig S1A and B). An indication for association of Nkp1 with Nkp2 in *S. cerevisiae* extracts has previously been reported (Brooks *et al*, 2010). We found that Nkp1-Nkp2 binds COMA, forming stable COMA-Nkp1-Nkp2 (Fig 1A). To understand the higher-order kinetochore architecture, we need to determine stoichiometries of kinetochore subassemblies. We used nanoflow mass spectrometry, to obtain information about reconstituted COMA-Nkp1-Nkp2 composition. In our nanoflow mass spectra of COMA-Nkp1-Nkp2 or COMA, we observed dimers of COMA-Nkp1-Nkp2 or COMA, respectively (Fig 1B and C), which were more abundant when we analysed these assemblies at higher micromolar concentrations. In related experiments, we found that COMA bound with a four-protein core variant of MIND (see Dimitrova *et al*, 2016), the orthologue of the human MIS12 assembly, forms prominent COMA-MIND dimers (Fig EV1A).

In our mass spectra of COMA, COMA was predominantly monomeric when we had analysed it at ~2 μM concentration (Fig 1C). At concentrations of ~3 μM or ~6 μM, a COMA dimer became more apparent (Fig 1C). These observations are consistent with our sedimentation-equilibrium analytical ultracentrifugation data of COMA (Appendix Fig S1C) that indicate a monomer–dimer equilibrium in solution in the concentration range of ~3–8 μM (for COMA monomer); and our static light scattering measurements—COMA loaded at higher concentrations (~43 μM—for dimeric COMA) on a size-exclusion chromatography column, elutes as a dimer (Fig 1D). To obtain information about COMA subunit topology, we used nanoflow mass spectrometry in tandem mode with collision-induced dissociation of protein assemblies in the gas phase. In our mass spectra, we observed dimeric COMA subassemblies without Ctf19, Mcm21 or Ctf19-Mcm21, but none without Ame1-Okp1 (Fig EV1B). These observations are consistent with our sedimentation-equilibrium ultracentrifugation data (Appendix Fig S1D), and our previous observation (Schmitzberger & Harrison, 2012) that Ctf19-Mcm21—on its own—is monomeric. We conclude that COMA dimerizes through Ame1-Okp1.

## Subunit arrangement in COMA-Nkp1-Nkp2

In our nanoflow mass spectra of COMA-Nkp1-Nkp2, we found that Nkp1 and Nkp2 associate separately with COMA (Fig EV2A and B), implying that both directly contact COMA. We did not find assemblies that contain Ctf19 or Mcm21 and Nkp1 or Nkp2, in the absence of Ame1-Okp1 or Okp1, suggesting that Ctf19-Mcm21 does not contact Nkp1-Nkp2 and that Ame1-Okp1 is the principal binding

**Figure 1.  Molecular composition of reconstituted *K. lactis* COMA-Nkp1-Nkp2 or reconstituted *K. lactis* COMA.**

A    (Left) Representative size-exclusion chromatography (SEC) chromatogram with absorbance measured at nanometres (nm) 260 or 280 (for this chromatogram and the following ones, absorbances are in units of 1,000[−1] (mAU)) of reconstituted *K. lactis* COMA-Nkp1-Nkp2. (Right) Image of Coomassie Blue-stained SDS–PAGE gel with fractions from principal SEC peaks. Molecular masses of protein standards (Mm. S.) are in kiloDalton (kDa); L: sample loaded on column. †: *Escherichia coli* DnaK; ╪: proteolysed Nkp1-Nkp2; Nkp1-Nkp2 is less prone to spontaneous proteolysis when associated with COMA.

B, C   Nanoflow mass spectra of COMA-Nkp1-Nkp2 (B) or COMA (C), acquired with different injected concentrations, showing an increase in COMA-Nkp1-Nkp2 dimers or COMA dimers with increasing injectant concentration. For this spectrum, and our other spectra of this type, the charge state of an assigned mass is indicated above its spectral peak.

D    Graph with measured differential refractive index (relative scale) and molar mass calculated with multi-angle light scattering (MALS) data of COMA eluting from SEC column (SEC-MALS). We injected COMA at a concentration of ~ 43 μM (COMA dimer) on the column. Dashed lines indicate expected molar masses for monomeric COMA or dimeric COMA: 142,337 g/mol or 284,674 g/mol; our experimentally determined molar mass from the principal SEC elution peak: 299,100 g/mol ± 5,683 g/mol (mean ± standard deviation from a single SEC-MALS experiment); max. dimeric COMA concentration, from refractive index measurement, in principal peak: 2.4 μM.

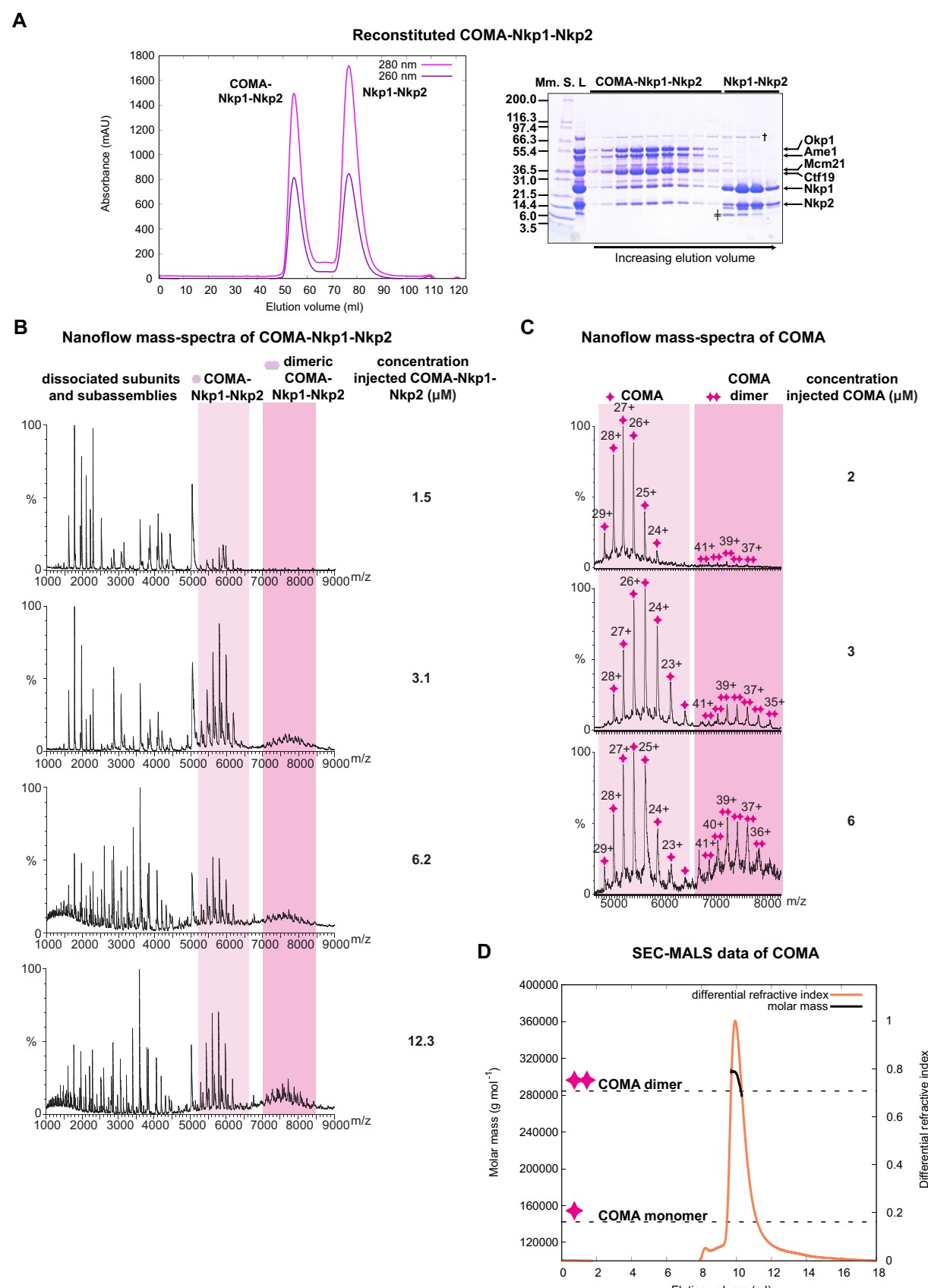

**Figure 1.**

partner for Nkp1-Nkp2. We confirmed this suggestion by size-exclusion chromatography (Fig EV3A and B). We conclude that Okp1-Ame1 binds Nkp1-Nkp2.

In spectra of tandem mass spectrometry of COMA, after its partial disruption with acetic acid in solution, we detected molecular masses corresponding to heterodimeric and heterotrimeric COMA subassemblies with all potential combinations of subunits, with the exception of those of Ame1 bound with Ctf19, Mcm21 or Ctf19-Mcm21, in the absence of Okp1 (Fig 2A and B, and Appendix Fig S2A and B). Our data suggest that in COMA, Okp1 contacts all other subunits and Ame1 interacts with Okp1 only.

### Flexible elements and structured segments in COMA proteins

Dynamic light scattering data that we measured of dimeric COMA (at ~21 μM concentration—assuming a COMA dimer) indicate that it has a hydrodynamic radius of ~12.9 nm (Appendix Fig S2C).

In our electron micrographs of negatively stained COMA (Appendix Fig S2D), however, we did not find distinctly shaped, globular particles, suggesting that, in the absence of other interacting macromolecules, COMA has flexible and/or unstructured elements.

To identify flexible elements and structured segments of COMA in solution, we incubated COMA and COMA-Nkp1-Nkp2 with deuterium oxide, and analysed mass spectra of their pepsin-proteolysed peptides. The deuterium-exchange patterns of Ctf19 or Mcm21 do not differ substantially between COMA and COMA-Nkp1-Nkp2 (Appendix Fig S3A and B), as we expected from our nanoflow mass spectra and size-exclusion chromatography experiments that we describe above.

For Okp1 in COMA, we found that more than half of its 383 residues exchanged rapidly with deuterium (Fig 3A; Appendix Fig S3C). Okp1 N-terminal regions (residues 1–164) and Okp1 C-terminal regions (330–383) are lacking a stable hydrogen-bonding network.

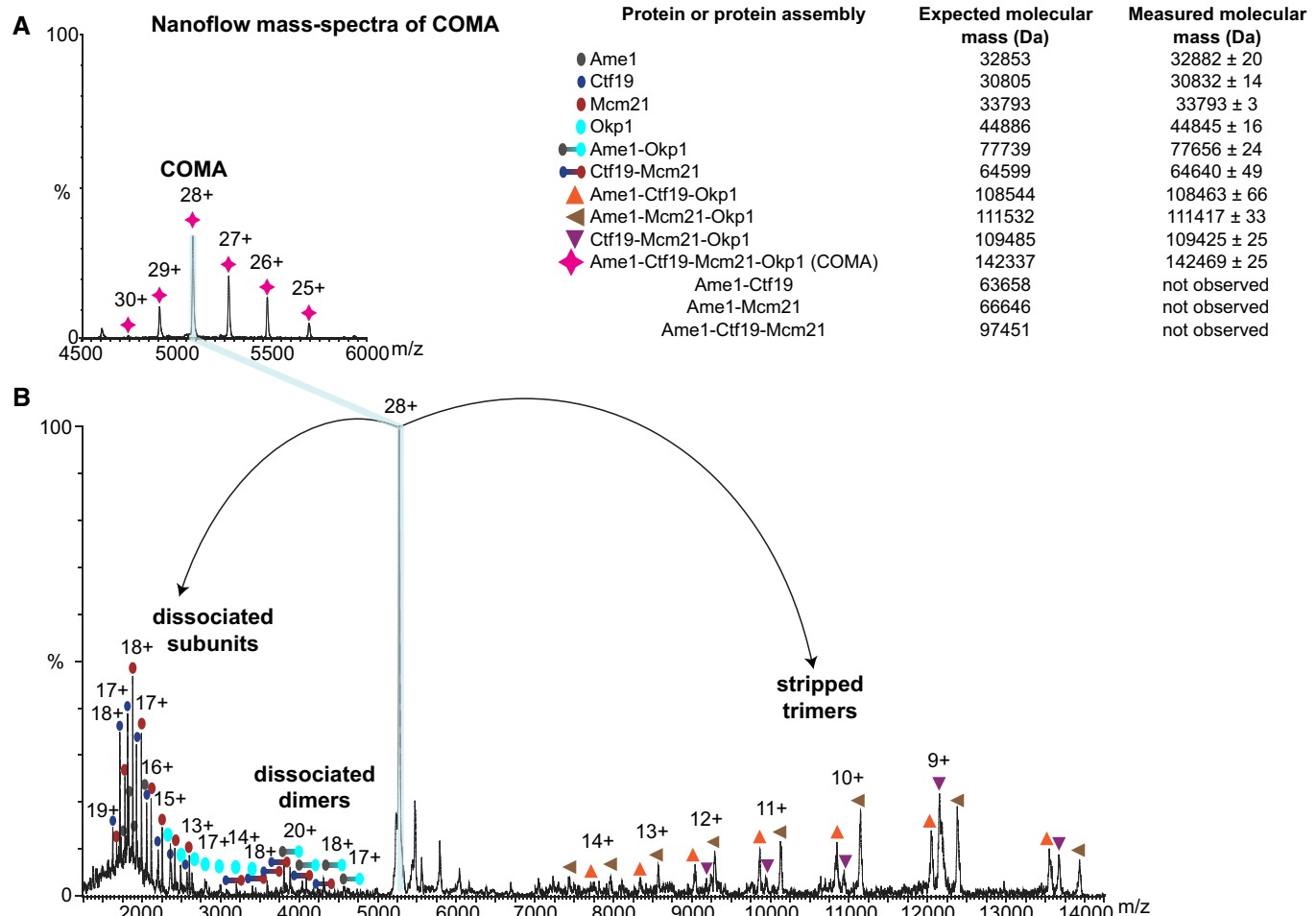

**Figure 2. Subunit topology of COMA.**

A   Excerpt from a representative nanoflow mass spectrum of COMA (full spectrum and enlarged spectrum areas are in Appendix Fig S2A). Inset table (on the right) shows identified proteins or protein assemblies, their measured mass values as mean ± standard deviation from multiple peaks in a peak series assigned to the same molecular species; and their respective expected mass.

B   Mass spectrum from tandem mass spectrometry of the isolated 28+ charge state of COMA (from spectrum shown in A), after its collision-induced dissociation in the gas phase.

Our data suggest that, in the absence of other interacting macromolecules, these regions are flexible. The pronounced protease sensitivity of the Okp1 N-terminal regions that we observed with our limited proteolysis experiments (see Figs EV4 and EV5; Tables EV1 and EV2) supports this suggestion. Okp1 parts that are protected the most from deuterium exchange in COMA are "core" (residues 166–211), "segment 2" (residues 234–264) and "segment 1" (residues 321–329). In COMA-Nkp1-Nkp2, part of the Okp1 C-terminus—"segment 3"—is protected (Fig 3A), which suggests that segment 3 binds Nkp1-Nkp2. We conclude that Okp1 has multiple structured segments, which are separated by flexible elements.

To assess the relevance of the different structured Okp1 segments for cell viability, we replaced in diploid *S. cerevisiae* cells one of the two native *Okp1* genes with a gene construct encoding either full-length Okp1 (Okp1_fl) or an Okp1 variant, which lacks one of the structured segments, with C-terminal flag epitopes (e.g. Okp1_fl-6×flag). Our Okp1 variants lacked either most of core (*S. cerevisiae* Okp1 residues 162–189), segment 2 (residues 236–265), segment 1 (residues 325–337) or segment 3 (residues 353–400; for sequence position, see Fig EV5). We evaluated viability of haploid spores after tetrad dissection. Tetrads of clones that encoded Okp1 versions without either core or segment 2 usually gave rise to only two viable haploid spores (out of four), which encoded native Okp1 (Appendix Fig S3D). Our isolated haploid spores that encoded Okp1_fl (*Okp1_fl*), or Okp1 without segment 1, or Okp1 without segment 3 were viable and grew similarly on solid standard growth medium (Appendix Fig S3D). We conclude that core and segment 2 are essential for viability, and that segment 1 and segment 3 are not essential for viability.

Deuterium-exchange patterns of Ame1 in COMA suggest that it has a central structured core, "Ame1 core" (residues 129–247; Fig 3B; Appendix Fig S3E and F), with N- and C-terminal elements, which, in the absence of other interacting macromolecules, are mostly flexible. We found with our limited proteolysis experiments that the Ame1 N-terminal region was particularly protease sensitive (Fig EV4; Appendix Fig S3F; Tables EV1 and EV2). In COMA-Nkp1-Nkp2, the C-terminal Ame1 "segment 1" (residues 268–292; Fig 3B) is more protected, which suggests that, like Okp1 segment 3, it interacts with Nkp1-Nkp2.

## Definition of COMA assembly requirements

To explore the relevance of structured segments and terminal regions in Ame1 and Okp1 for COMA formation, we reconstituted COMA variants with truncations of Ame1 and Okp1. The strongly associating D-RWD domains of Ctf19 and Mcm21 suffice for stable assembly with full-length Ame1-Okp1, while the N-terminal 106 residues of Ctf19 or Mcm21 are dispensable for COMA assembly (Schmitzberger & Harrison, 2012). We co-expressed coding regions for polyhistidine-tagged Mcm21 D-RWD domain (His-Mcm21$_{D-RWD}$) and Ctf19 D-RWD domain (Ctf19$_{D-RWD}$), with coding regions for Ame1 truncations and Okp1 truncations (Fig 4A); screened for soluble protein production; and purified assemblies with affinity chromatography (Fig 4B). We found that N-terminal and C-terminal regions of Okp1 (residues 1–122 and 337–383) and N-terminal and C-terminal regions of Ame1 (residues 1–113 and 226–292) are dispensable for the formation of stable COMA variants. Minimized COMA variants form with Ame1 core, Okp1 core, Okp1 segments 1

and 2, and the Ctf19-Mcm21 D-RWD domain modules (Fig 4B). We did not observe reconstitution of COMA variants with Okp1 versions that lack segment 1. We conclude that Okp1 segment 1 is required for association of Okp1-Ame1 with Ctf19-Mcm21.

With a reconstitution strategy analogous to that for COMA, we found that Ame1-Okp1 variants are stable in the absence of Ctf19-Mcm21 (Fig 4C; Appendix Fig S4). Ame1-Okp1 forms with Ame1 core, and Okp1 core and segment 2. Ame1 core and Okp1 segment 2 are predicted to be coiled coils, which suggests an Ame1-Okp1 coiled coil. In contrast to COMA assembly, Ame1-Okp1 assembly does not require Okp1 segment 1 (Fig 4C). Okp1 segment 1 is selectively required for Ctf19-Mcm21 binding.

Consistent with this conclusion, we were unable to reconstitute Ctf19$_{D-RWD}$-Mcm21$_{D-RWD}$-Okp1 assemblies with Okp1 constructs that lack segment 1, but we were able to reconstitute Ctf19$_{D-RWD}$-Mcm21$_{D-RWD}$-Okp1 assemblies with Okp1 constructs spanning core and segments 1 and 2 (Fig 4D). We found that these Okp1 variants were prone to spontaneous proteolysis. In mass spectra of our purified Ctf19-Mcm21-Okp1 variants (with Okp1$_{106-336}$ or Okp1$_{123-336}$ (subscript denotes residue numbers)), prominent Okp1 fragments that had co-purified with Ctf19$_{D-RWD}$-Mcm21$_{D-RWD}$ included segments 1 and 2, but lacked core (one such fragment included residues 234–336). Our observation suggested that Okp1 core is dispensable for binding Ctf19-Mcm21. We found indeed that a recombinant Okp1 variant (residues 229–336) that includes segments 1 and 2, but lacks core, binds Ctf19-Mcm21 (Fig 5A). This variant was less prone to spontaneous proteolysis than our above-described Okp1 variants. We conclude that Okp1 binding to Ctf19-Mcm21 does not require Okp1 core.

## Ame1-Okp1 C-termini bind Nkp1-Nkp2

To test our suggestion, from deuterium-exchange data (Fig 3A), that Okp1 segment 3 binds Nkp1-Nkp2, we combined our less proteolysis-prone Ctf19-Mcm21-Okp1 variant without segment 3, or a similar variant that included segment 3, with Nkp1-Nkp2, and analysed these combinations with size-exclusion chromatography. Our Ctf19$_{D-RWD}$-Mcm21$_{D-RWD}$-Okp1 variant without Okp1 segment 3 did not co-elute with Nkp1-Nkp2 (Fig 5A), but our Ctf19$_{D-RWD}$-Mcm21$_{D-RWD}$-Okp1 variant that includes Okp1 segment 3 co-eluted with Nkp1-Nkp2 (Fig 5B). Okp1 segment 3 binds Nkp1-Nkp2 to COMA. To investigate the relevance of Okp1 segment 3 for recruiting Nkp1-Nkp2 to centromeres in living cells, we genetically modified our haploid *S. cerevisiae* clones with Okp1_fl (*Okp1_fl*) or with Okp1 that lacks segment 3. We modified these clones so that they encode green fluorescent protein (GFP) fused to the C-terminus of either Nkp1 or Nkp2. In living mitotically cycling cells of such clones, we monitored GFP signals with fluorescence microscopy. In *Okp1_fl* cells—budded or non-budded, Nkp1-GFP and Nkp2-GFP signals emanated from distinct foci that mark kinetochore clusters (Fig 5C). In cells without Okp1 segment 3, only residual fluorescence of Nkp1-GFP or Nkp2-GFP was visible (Fig 5C; more of Nkp2-GFP than of Nkp1-GFP). We conclude that in living cells with Okp1 that lacks segment 3 (*Okp1_nnΔ*), centromere localization of Nkp1-Nkp2 is abrogated. Segment 3 is required for localization of Nkp1 and Nkp2 to centromeres.

The protection from deuterium exchange and reduced protease sensitivity that we observe for Ame1 segment 1 in COMA-Nkp1-Nkp2

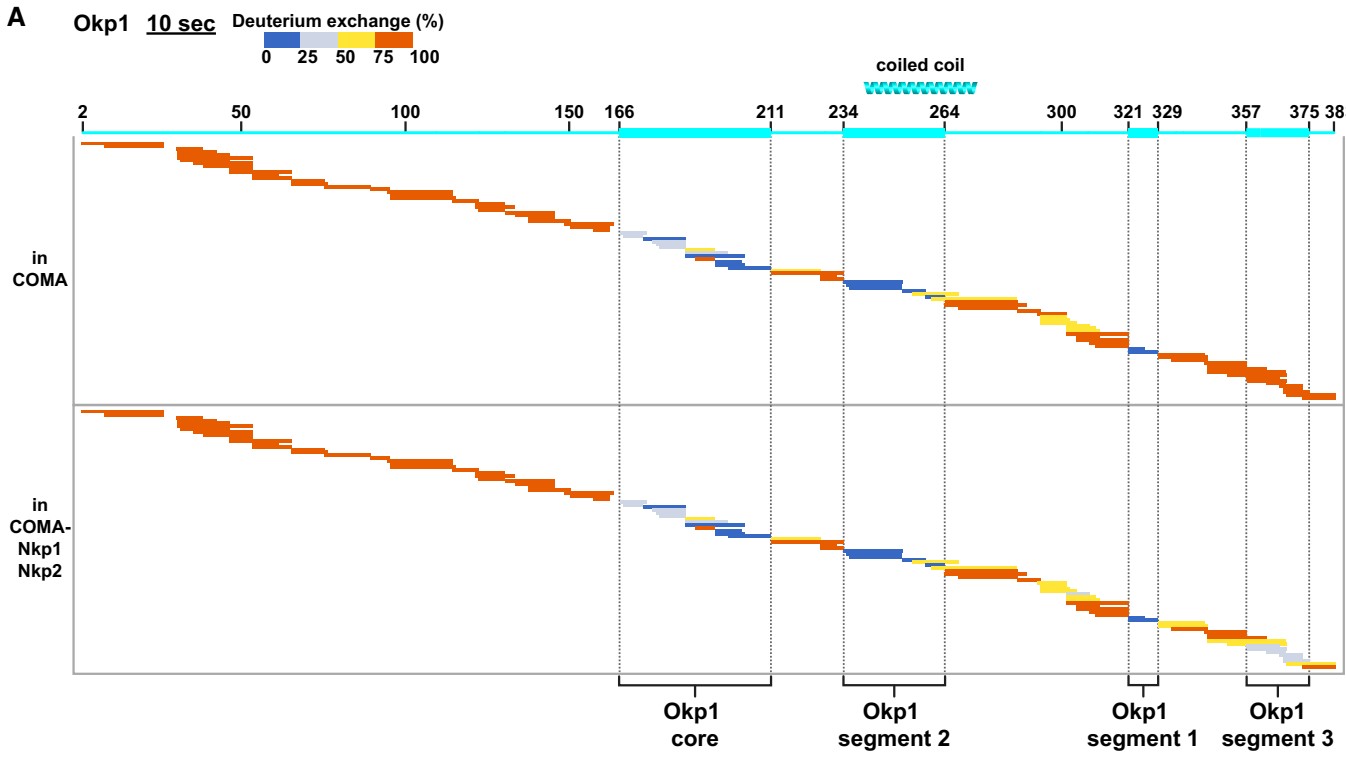

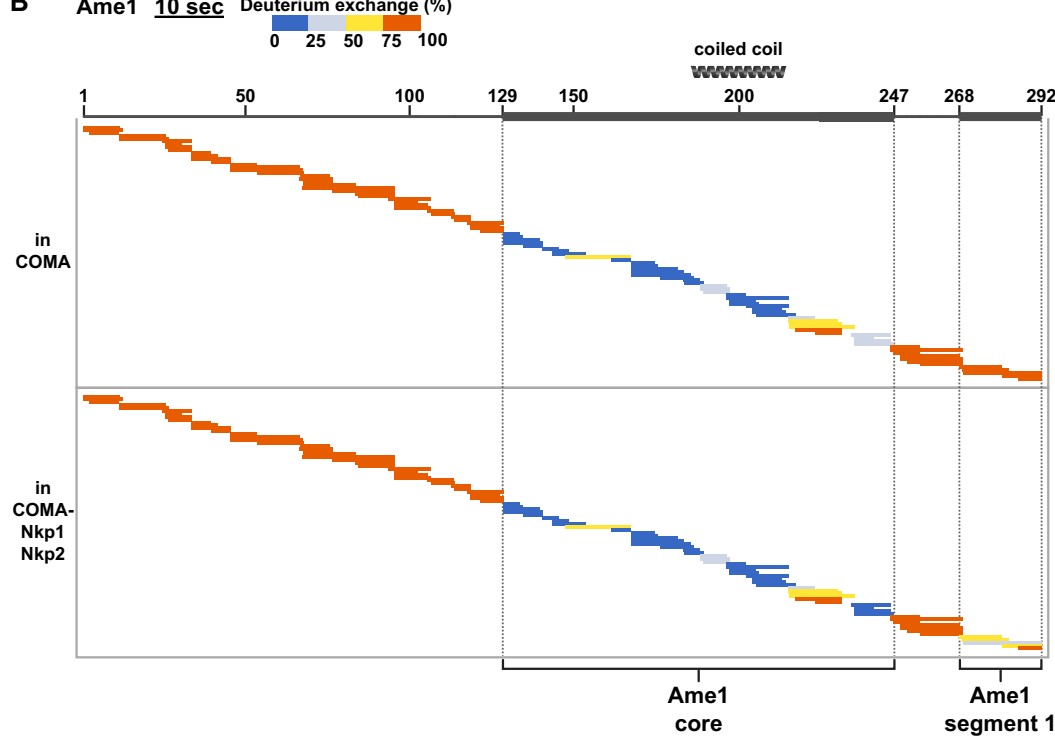

**Figure 3.** **Comparative deuterium-exchange experiments identify structured segments and flexible elements of Okp1 or Ame1 in COMA-Nkp1-Nkp2.**

A, B    Plots showing hydrogen-deuterium-exchanged peptides, after 10 s of deuterium exchange of Okp1 (A) or Ame1 (B), each in COMA or COMA-Nkp1-Nkp2. For this representation, and all following of this type, detected peptides are colour-coded according to their measured deuterium exchange in % of the maximum measured exchange (1–25%: dark blue; 26–50%: light blue; 51–75%: yellow; 76–100%: red). Peptides are represented as bars. Each bar is plotted row-wise corresponding to its position in the amino acid sequence of the protein it derives from. Amino acid sequence scheme is illustrated at the top with thin or thick lines; thin line: flexible element; thick line: structured segment. Positions of predicted coiled coils are shown as helices. Plots for full time courses are in Appendix Fig S3C and E.

Source data are available online for this figure.

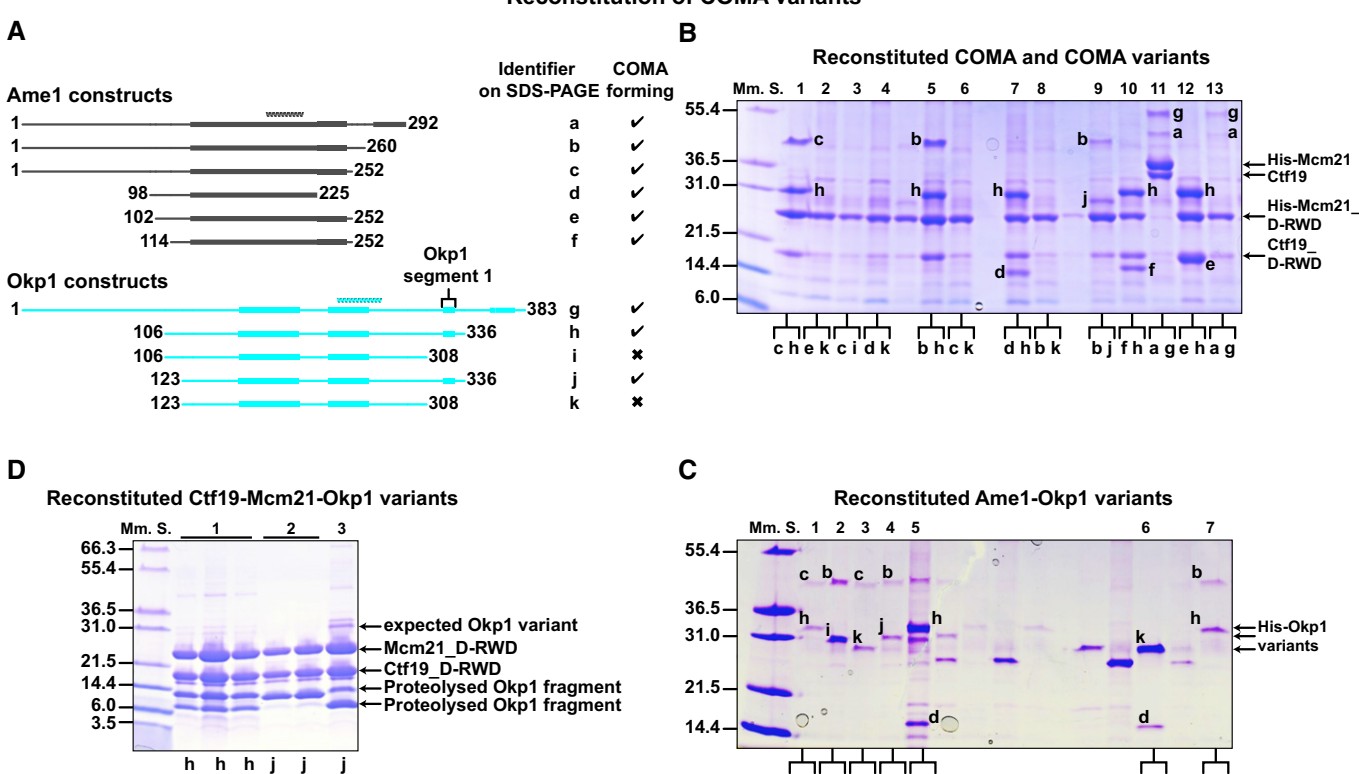

**Figure 4. Molecular requirements for COMA assembly.**

A  Schematics of our designed Ame1 and Okp1 constructs; our identified structured segments and coiled coils are as we show in Fig 3.

B, C  Representative images of Coomassie Blue-stained SDS–PAGE gels with Ni²⁺-affinity purification eluates of (B) full-length COMA (lane 11) with full-length polyhistidine-tagged Mcm21 (His-Mcm21) and Ctf19, or COMA variants with polyhistidine-tagged Mcm21 D-RWD domain (His-Mcm21_D-RWD; *K. lactis* Mcm21 residues 108–293) and Ctf19 D-RWD domain (Ctf19_D-RWD; *K. lactis* Ctf19 residues 107–270); of (C) Ame1-Okp1 variants with polyhistidine-tagged Okp1 variants (His-Okp1 variants). We show one-letter identifiers for protein versions in (A). Lanes between lanes 5 and 6 in (C) are not relevant.

D  Image of SDS–PAGE gel with fractions of two purified Ctf19-Mcm21-Okp1 variants (after SEC); expected migrating position of non-proteolysed Okp1 variant (compare with its position in SDS–PAGE images of COMA or Ame1-Okp1 variants in (B or C)) is marked; 2 and 3: SEC-purified fractions of samples from different elution fractions from ion-exchange chromatography. When we co-expressed full-length *Okp1* with *Ctf19* and *Mcm21*, which has a coding region for an N-terminal polyhistidine tag, we did not co-purify substantial amounts of full-length Okp1 with Ctf19-Mcm21, probably because Okp1 was prone to proteolysis (in the absence of Ame1).

(Figs 3B and EV4) indicates that, like the Okp1 C-terminus, the Ame1 C-terminus interacts with Nkp1-Nkp2. Through analysis of our nanoflow mass spectra, we found that Nkp1 and Nkp2 separately contact Ame1-Okp1 (Fig EV2B), suggesting distinct binding sites in COMA for Nkp1 and Nkp2.

Nkp1 and Nkp2 both have unstructured C-terminal parts in Nkp1-Nkp2 alone that are protected from deuterium exchange in COMA-Nkp1-Nkp2 (Fig 5D and E; Tables EV1–EV3; Appendix Fig S5A–D). In Nkp1, more than 60 residues change protection, in Nkp2~45 residues—large regions, relative to their respective (residue) sizes. We conclude that Nkp1 C-terminal part and Nkp2 C-terminal part interact extensively with Ame1-Okp1. Comparison of the deuterium-exchange patterns of Nkp1 on its own with those in Nkp1-Nkp2 (Appendix Fig S5E) suggests that the N-terminal part of Nkp1 binds Nkp2. Our reconstitution experiments (see Appendix Fig S5F) support this suggestion. Since the Nkp2 C--terminal part contacts Ame1-Okp1, the Nkp2 N-terminal part presumably contacts Nkp1.

### Molecular basis for Okp1 interactions in COMA

To resolve interactions of Okp1 segments that we had identified, with our reconstitution experiments (Fig 4), as relevant for Ame1 or Ctf19-Mcm21 binding in greater detail, we used deuterium exchange (Appendix Fig S6A–E). Comparison of the exchange patterns of Okp1 peptides from full-length COMA with those from an Ame1-Okp1 variant or from a Ctf19-Mcm21-Okp1 variant (Fig 6A) allowed us to detect differences in Okp1, if either Ctf19-Mcm21 or Ame1 was absent. With Ame1 bound, Okp1 segment 2 is protected from exchange; in the absence of Ame1, it is unprotected (Fig 6A). We conclude that segment 2 binds Ame1 core.

In the presence of Ctf19-Mcm21, Okp1 segment 1 is protected from exchange (Fig 6A; Appendix Fig S6A and C); in the absence of Ctf19-Mcm21, it is disordered. Our observation confirms the requirement of Okp1 segment 1 for Ctf19-Mcm21 binding. To define the corresponding contacts for Okp1 in the Ctf19-Mcm21 D-RWD domain structure, we compared the deuterium-exchange patterns of

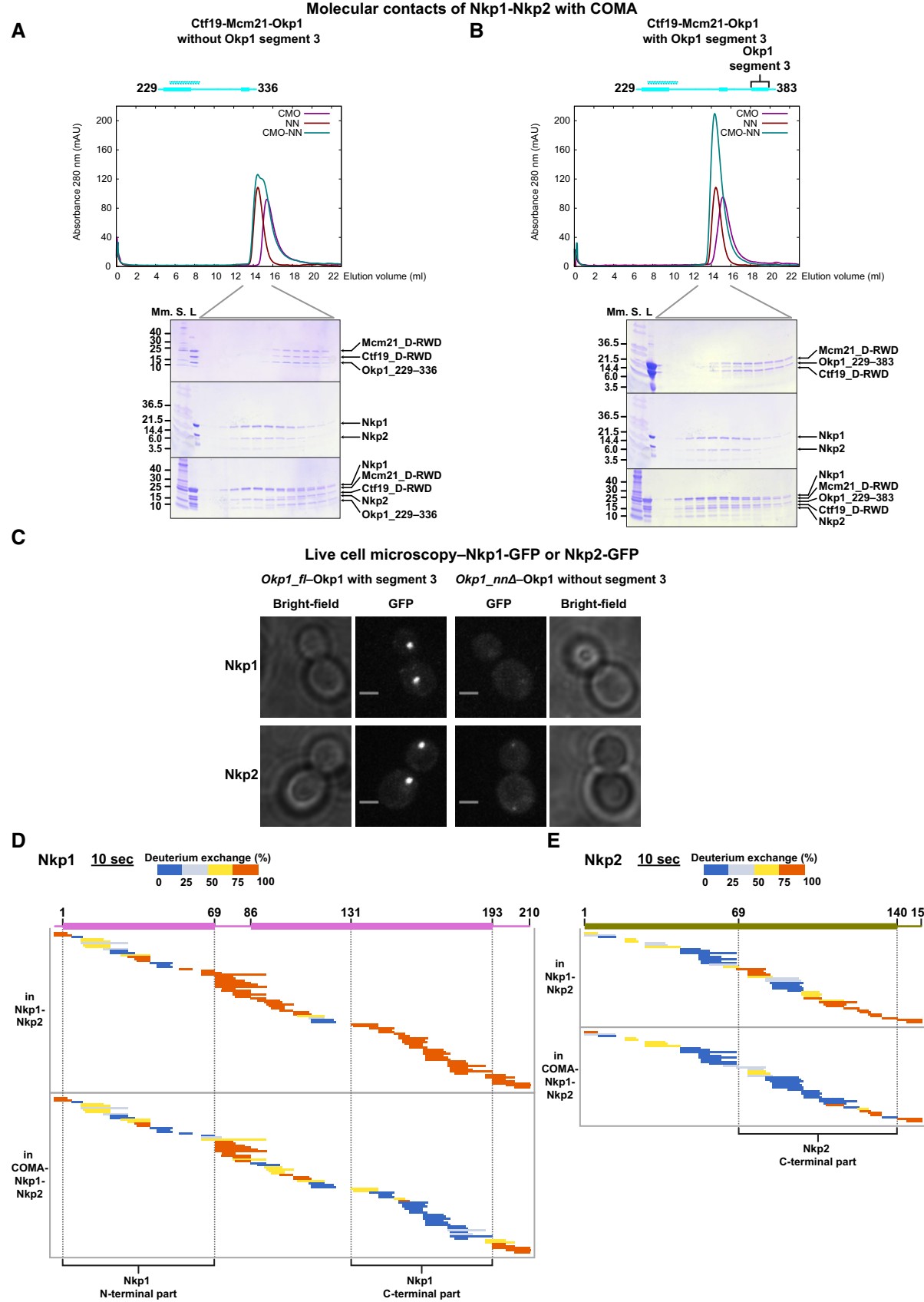

**Figure 5.**

◄

**Figure 5.　Interactions of Okp1 or Ame1 with Nkp1-Nkp2.**

A, B　Overlayed chromatograms showing absorbance at 280 nm from SEC of Ctf19-Mcm21-Okp1 variants without Okp1 segment 3 (A) or with Okp1 segment 3 (B); Ctf19-Mcm21-Okp1 variant: CMO, Nkp1-Nkp2: NN, Ctf19-Mcm21-Okp1 variant combined with Nkp1-Nkp2: CMO-NN; and images of SDS–PAGE gels with equivalent elution fractions from five SEC experiments; L: loaded on column. The elution volume range that we analysed fractions of on SDS–PAGE is indicated below the chromatograms.

C　Representative microscopy images of living haploid *S. cerevisiae* cells with *Okp1_fl* (integrated full-length *Okp1*) or our *Okp1* version that lacks the coding region for segment 3—*Okp1_nnΔ*, with Nkp1-GFP or Nkp2-GFP. Fluorescence images are maximum intensity projected, and we show those from *Okp1_fl* or *Okp1_nnΔ* with the same kinetochore protein–fluorescent protein fusion on the same intensity scale; scale bar: 2 μm.

D, E　Plots showing hydrogen-deuterium-exchanged peptides, after 10 s of deuterium exchange of Nkp1 (D) or Nkp2 (E), each in Nkp1-Nkp2 or COMA-Nkp1-Nkp2. Nkp1 has an N-terminal SNA residual. Plots for full time courses are in Appendix Fig S5A and B.

Source data are available online for this figure.

Ctf19 or Mcm21 in Ctf19-Mcm21 with those of Ctf19 or Mcm21 in COMA (Fig 6B; Appendix Fig S6D and E). The deuterium-exchange patterns of Ctf19 or Mcm21 in Ctf19-Mcm21 are essentially consistent with our previously reported data on flexible regions and the globular structure of the D-RWD domains of Ctf19-Mcm21 (Schmitzberger & Harrison, 2012). Most of the N-terminal 100 residues of Ctf19 or Mcm21 exchange rapidly (except Ctf19 residues 38–56; see figure legend for Appendix Fig S6). For Mcm21, the exchange patterns in Ctf19-Mcm21 and COMA are similar; there are small differences only for peptides from its central α-helices (α2, α3; Fig 6B; Appendix Fig S6E). For Ctf19, we observe a dramatic difference for the hydrophobic, partially sequence-conserved C-terminal part of its C-terminal RWD domain (RWD-C). This part is unprotected in the absence of Okp1. With Okp1 bound, it is protected—even after long incubation periods with deuterium (Fig 6B; Appendix Fig S6D), suggesting a very strong association between Ctf19 and Okp1.

In our purified Ctf19$_{D-RWD}$-Mcm21$_{D-RWD}$-Okp1-variant samples, Okp1 variants with segments 1 and 2 (Okp1$_{229–336}$; Fig 5A), or with segments 1, 2 and 3 (Okp1$_{229–383}$; Fig 5B), spontaneously proteolysed, probably because their Ame1 binding site or Ame1 and Nkp1-Nkp2 binding sites were unstructured. Corresponding Okp1 fragments, which we identified in our mass spectra, had N-termini corresponding to residues between segments 1 and 2, and C-termini corresponding to residues at segment 3 N-terminus or at segment 1 C-terminus (see legend to Fig EV4). Guided by this observation, we designed an Okp1 variant (Okp1$_{295–360}$) that, in addition to lacking Okp1 core and segment 3, also lacks segment 2—the Ame1 binding site. Reconstituted with Ctf19$_{D-RWD}$-Mcm21$_{D-RWD}$, this Okp1 variant was proteolysis resistant, and we purified this minimized stable assembly homogeneously (Appendix Fig S7A). A similar minimized Ctf19-Mcm21-Okp1 variant that includes Okp1 segment 3 (Okp1$_{295–383}$) bound Nkp1-Nkp2 (Appendix Fig S7B), as we had anticipated. We conclude that Okp1 binding segments specific for Ctf19-Mcm21, Nkp1-Nkp2 or Ame1 are spatially separated.

### Structural determinants for Ctf19-Mcm21 binding to Okp1

The definition of a minimized stable Ctf19-Mcm21-Okp1 variant that includes Okp1 segment 1 (Appendix Fig S7A), as we describe above, was essential to obtain diffracting crystals. In contrast to crystals that we obtained of full-length Ame1-Okp1 bound with the Ctf19-Mcm21 D-RWD domains, which did not diffract to higher than 50 Å resolution, our minimized ternary assembly formed diffracting crystals. We determined its structure with X-ray diffraction data extending to 2.1 Å resolution (Table EV4). From our diffraction data, we modelled most of the Ctf19-Mcm21 D-RWD domains. We did not

observe substantial electron density for Okp1$_{295–360}$ residues 295–317 or 343–360 (Appendix Fig S7C). Since we did not find indications that Okp1$_{295–360}$ proteolysed in our crystals (Appendix Fig S7D), we conclude that these residues are flexible. This conclusion is consistent with our deuterium-exchange data (Figs 3A and 6A). Only Okp1 residues 318–342 are structured in our crystals (Fig 6C and D; Movie EV1). Of these, residues 338–342 do not interact substantially with Ctf19-Mcm21. They interact with residues 318–342 of another Okp1 molecule—related by non-crystallographic twofold rotational symmetry. Because residues 338–342 are not conserved among budding yeasts (Fig EV5), we assume their interaction in our crystals is a result of crystallization, rather than specific for COMA dimerization.

Okp1 residues 319–337 interact most extensively with Ctf19 (~850 Å$^2$ of buried surface area; Fig 6C; Movie EV1). Okp1$_{319–337}$ is crescent shaped and wraps around the penultimate α-helix (α5) of Ctf19 RWD-C—the area most protected from deuterium exchange with bound Okp1 (Fig 6B). Okp1 segment 1 is the main binding site for Ctf19-Mcm21, as we anticipated from our biochemical characterization. Segment 1 is α-helical and lies in the cleft between Ctf19 and Mcm21, at the intersection of their D-RWD domains (Fig 6C). This position protects it effectively from proteases. Okp1 segment 1 is followed by a short β-strand-like segment (residues 330–332) that wedges in the Ctf19 hydrophobic groove between α5 and β8, augmenting the β-sheet of Ctf19 RWD-C (Fig 6D; Appendix Fig S7E). Dipole–dipole interactions between the N-terminus of the Okp1 α-helix and the C-terminus of Ctf19 α5 stabilize Ctf19-Okp1. Specific Ctf19-Okp1 interactions are of Okp1 Phe329 (a residue with a bulky hydrophobic side chain in this position is present among many budding yeasts; Figs 6E and EV5), with residues in the hydrophobic groove of Ctf19 RWD-C and of Okp1 Asn330, which is at a distinctive kink in the structure right after segment 1 (Fig 6D) and conserved among budding yeasts, with the Ctf19 Pro236 oxygen. A proline in the equivalent sequence position of *K. lactis* Ctf19 Pro236 is conserved among Ctf19 proteins and orthologous mammalian CENP-P proteins (see supplementary Fig S6B in ref. Schmitzberger & Harrison, 2012). In the GCN1 RWD domain, the region that is structurally equivalent to the Ctf19 loop with Pro236 was shown to be important for folding and stability (Nameki *et al*, 2004). Our data (also see figure legend to Appendix Fig S8) suggest that Okp1 binding stabilizes Ctf19. Okp1 segment 1 side chains also contact side chains in Mcm21 α2 and α3. We observe the following interaction pairs: Mcm21_Lys189—Okp1_Glu320, Mcm21_Lys200—Okp1_Asp328 and Mcm21_Lys207—Okp1_Asn330 (Fig 6D). These contacts are consistent with small mass differences, of 1 or 2 Da, for deuterium-exchanged peptides from Mcm21 residues 190–207 in

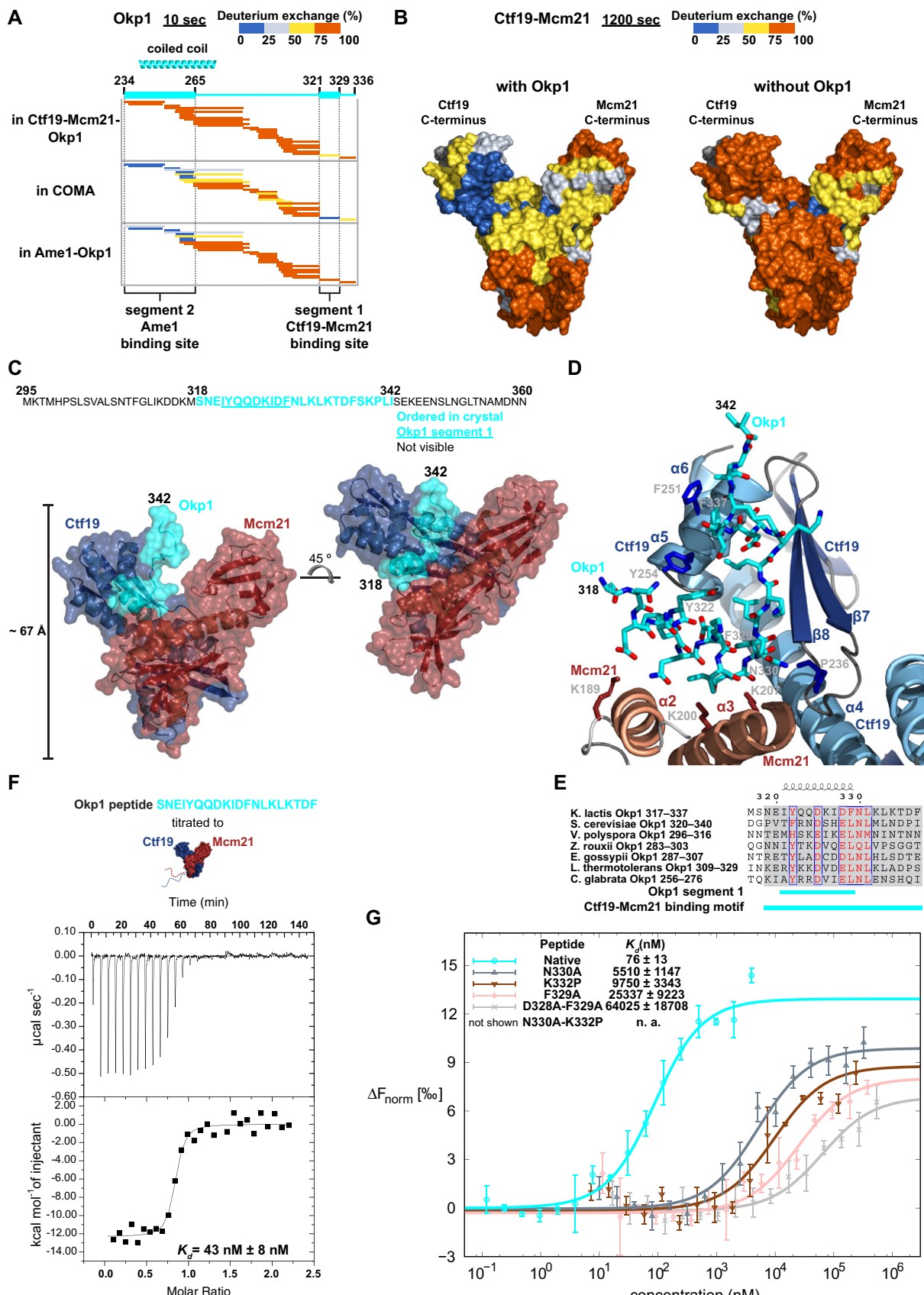

**Figure 6.**

◄

**Figure 6.   Molecular structural characteristics of the Ctf19-Mcm21 binding motif of Okp1.**

A   Plots showing deuterium-exchanged peptides, after 10 s of deuterium exchange of Okp1 in Ctf19$_{D-RWD}$-Mcm21$_{D-RWD}$-Okp1$_{229-336}$, in full-length COMA, or in Ame1$_{1-260}$-Okp1$_{123-336}$. Plots for full time course are in Appendix Fig S6C.

B   Deuterium exchange of peptides, from deuterium-exchange data after 1,200 s of deuterium exchange of Ctf19 and Mcm21, mapped onto the surface of the structure of the Ctf19-Mcm21 D-RWD domains (PDB code: 3ZXU); left: Ctf19-Mcm21 in COMA; right: Ctf19-Mcm21 alone. Residues without deuterium-exchange data are coloured dark grey. Plots for full time courses are in Appendix Fig S6D and E.

C   Semi-transparent surface representation of our Ctf19$_{D-RWD}$-Mcm21$_{D-RWD}$-Okp1$_{295-360}$ crystal structure, with secondary structure cartoon diagram underneath; left: in a similar orientation as Ctf19-Mcm21 in (B). In Okp1 letter-sequence, turquoise coloured letters are Okp1 residues that are ordered in our crystal structure; underlined letters are in segment 1; black letters are disordered residues.

D   Close-up view of the Okp1 binding site of Ctf19-Mcm21 with Ctf19-Mcm21 binding motif of Okp1; Okp1 residues, and some Ctf19 or Mcm21 side chains, are shown as sticks (nitrogen: blue, oxygen: red); Ctf19-Mcm21 shown as secondary structure cartoon diagram.

E   Sequence alignment of budding yeasts' Ctf19-Mcm21 binding motif of Okp1. Similar residues are coloured red. We show full alignment in Fig EV5.

F   Representative baseline-subtracted raw isothermal calorimetry data and derived binding isotherm from titration of Okp1-derived synthetic peptide, with the Ctf19-Mcm21 binding motif, to full-length Ctf19-Mcm21. Mean value for dissociation constant ($K_d$) and its standard error of regression are from three independent experiments.

G   Plots from microscale thermophoresis measurements with fluorescently labelled full-length Ctf19-Mcm21 and with Okp1-derived synthetic peptide (same as in F), or with Okp1 peptide variants that differ in one or two residue positions from the native Okp1 sequence (e.g. N330→A: N330A). Normalized fluorescence, from which the fluorescence of unbound Ctf19-Mcm21 was subtracted ($\Delta F_{norm}$), is plotted against peptide concentration (logarithmic scale). Error bars in plot show standard deviation from mean value, from three independent measurements. For dissociation constant ($K_d$), we calculated mean value and confidence interval (68% probability that $K_d$ is within given range) with data from three independent experiments. Variant N330A-K332P had no detectable binding in the peptide concentration range that we measured.

Source data are available online for this figure.

mass spectra of Ctf19-Mcm21 and those of COMA (Appendix Fig S6E). These residues are more protected with Okp1 bound. Binding to Ctf19-Mcm21 stabilizes the Ctf19-Mcm21 binding motif of Okp1 (Fig 6E), because in the absence of Ctf19-Mcm21, segment 1 is disordered (Fig 6A).

Our Ctf19-Mcm21-Okp1 structure is in excellent agreement with our deuterium-exchange data. We conclude that the hydrophobic groove in Ctf19 RWD-C and Mcm21 α2 and α3 are the principal binding sites for Okp1 in Ctf19-Mcm21.

## Specificity of the Ctf19-Mcm21 binding motif

To determine the affinity of our structurally defined Ctf19-Mcm21 binding motif in Okp1 for Ctf19-Mcm21 in solution, we measured, with isothermal titration calorimetry or microscale thermophoresis, association of a synthetic Okp1-derived peptide that includes the Ctf19-Mcm21 binding motif (Okp1 residues 318–337; Fig 6E) with full-length Ctf19-Mcm21. We derived a dissociation constant ($K_d$) of ~40 nM or ~80 nM (Fig 6F and G). The high binding affinity is consistent with the pronounced deuterium-exchange protection of Ctf19 RWD-C and Okp1 segment 1 (Fig 6A and B). We also measured dissociation constants with Okp1-variant peptides that differed from the native Ctf19-Mcm21 binding motif in one or two residues (Fig 6E) that make important contacts with Ctf19-Mcm21 in our structure. Such variants have a dissociation constant that is two or three orders of magnitude higher than that of the native binding motif (Fig 6G). We conclude that binding is sequence specific.

## A subset of inner kinetochore proteins depends on the Ctf19-Mcm21 binding motif for centromere localization

Confirming our results about the specificity of the Ctf19-Mcm21 binding motif, in affinity purifications of recombinant *K. lactis* COMA with an Okp1 variant that lacks most of this motif (residues 322–334; Okp1_cmΔ; Fig 7A), we did not observe substantial amounts of Ame1-Okp1_cmΔ co-purifying with Ctf19-Mcm21

(Appendix Fig S8A). To test the functional contribution of the Ctf19-Mcm21 binding motif for COMA assembly in yeast cells, we genetically modified our haploid *S. cerevisiae* clones that encode full-length Okp1 or our Okp1 variant that lacks segment 1 (residues 325–337)—most of the part that corresponds to the Ctf19-Mcm21 binding motif in *K. lactis* Okp1 (Fig 6E). We modified these clones, so that they encode myc epitopes fused to the Ctf19 C-terminus. From extracts of mitotic cells, Ctf19 co-immunoprecipitated with full-length Okp1 (Okp1_fl)—as we expected, but did not co-immunoprecipitate with Okp1 that lacks segment 1 (Okp1_cmΔ; Fig 7B; Appendix Fig S8B and C). We conclude that the Ctf19-Mcm21 binding motif in Okp1 is selectively required for native COMA assembly.

To investigate the relevance of the Ctf19-Mcm21 binding motif for overall kinetochore organization in living mitotically cycling cells, we monitored with fluorescence microscopy the centromere localization of GFP-tagged subunits from the inner kinetochore or outer kinetochore, in *S. cerevisiae* clones *Okp1_fl* or *Okp1_cmΔ*. Ctf19-GFP and Mcm21-GFP co-localized with red fluorescent protein (RFP)-tagged outer kinetochore protein Nuf2, which marks kinetochore clusters, in *Okp1_fl*, but did not co-localize in *Okp1_cmΔ* (Fig 7C), consistent with our biochemical analyses. We observed this effect in unbudded, small budded and large budded cells. Centromere localization of Ame1[CENP-U], and Mtw1[MIS12] and Nnf1[PMF1]—kinetochore subunits of the MIND assembly that depend on Ame1 for centromere localization, was not affected in *Okp1_cmΔ* (Fig 7D; Appendix Fig S8D). This observation is in agreement with the selectivity of the Ctf19-Mcm21 binding motif, which we had established with our biochemical experiments. The Mif2[CENP-C] GFP signal was, however, substantially higher in *Okp1_cmΔ* than in *Okp1_fl* (Fig 7D). Our observation is in general agreement with the observation that Mif2[CENP-C] binds Ame1[CENP-U]-Okp1[CENP-Q] (Hornung *et al*, 2014), but that it does not bind Ctf19[CENP-P]-Mcm21[CENP-O]. We found, with our experiments with *in vitro* translated Mif2, that COMA binds the central part of Mif2 (Fig EV6A).

With imaging experiments analogous to those that we describe above, we found that Chl4[CENP-N] or Iml3[CENP-L] did not specifically

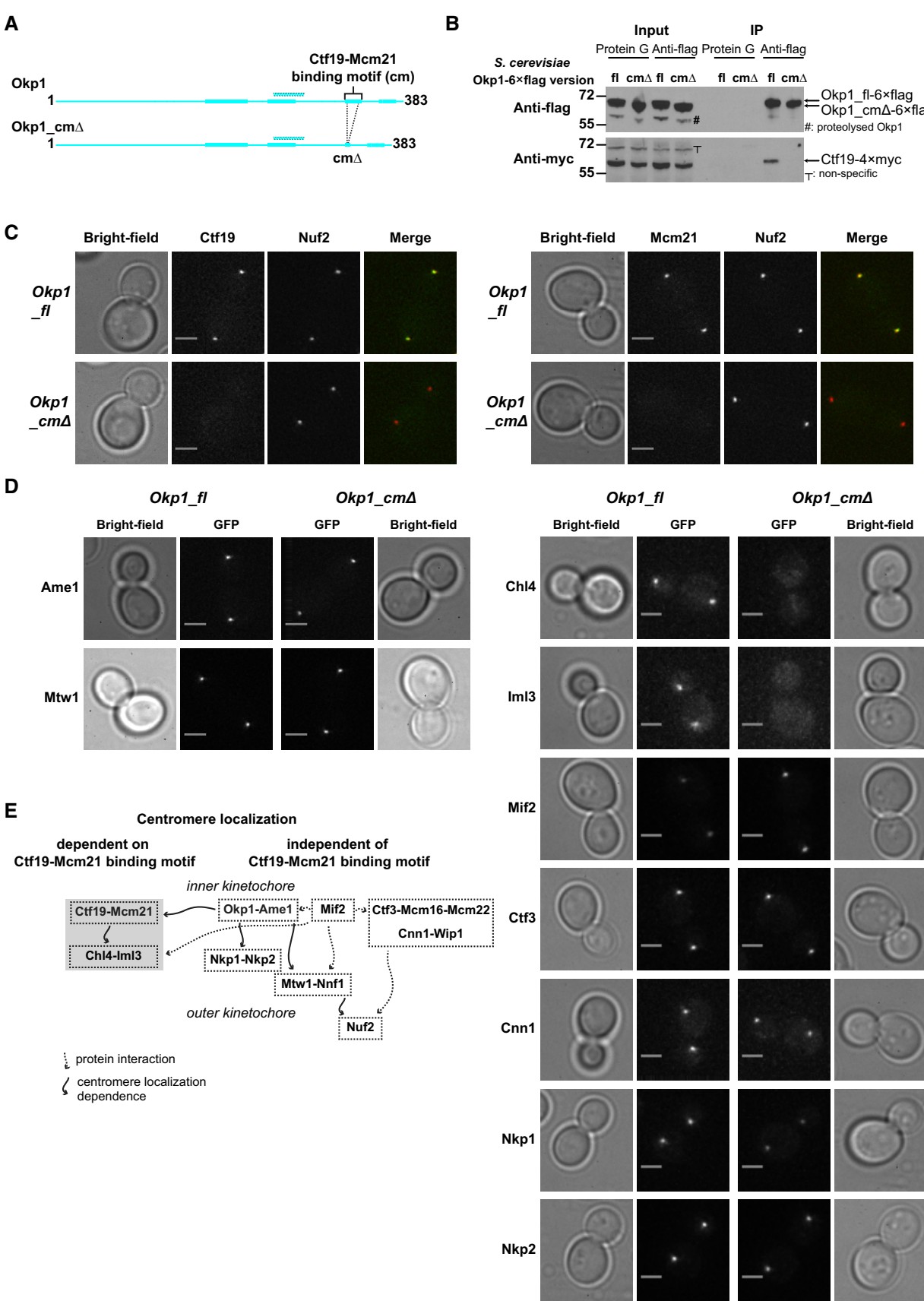

**Figure 7.**

◀

**Figure 7. Effect of absence of the Ctf19-Mcm21 binding motif on kinetochore organization.**

A  Schematic of *K. lactis* Okp1 or *K. lactis* Okp1 without the Ctf19-Mcm21 binding motif (Okp1_cmΔ); annotation is as shown in Fig 3A.
B  Representative images of Western blots of immunoprecipitated fractions from *S. cerevisiae* extracts with Okp1-6×flag and Ctf19-4×myc; fl: Okp1_fl; cmΔ: Okp1 variant without the Ctf19-Mcm21 binding motif (Okp1_cmΔ); IP: immunoprecipitated samples; Protein G: beads coated with Protein G only (used as control); positions of molecular masses of standards are indicated; uncropped images are in Appendix Fig S8B and C.
C  Representative microscopy images of living haploid *S. cerevisiae* cells with Nuf2-mCherry, and Ctf19-GFP (left panel) or Mcm21-GFP (right panel), either with *Okp1_fl* or with *Okp1_cmΔ*. We show merged green fluorescence signals and red fluorescence signals as pseudo-colours. We show fluorescence images of cells with *Okp1_fl* or *Okp1_cmΔ* with the same kinetochore protein–fluorescent protein fusion on the same intensity scale (also applies to images in D). Fluorescence images and those in (D) are deconvoluted and maximum intensity projected; scale bar: 2 μm.
D  Representative microscopy images of living haploid *S. cerevisiae* cells with *Okp1_fl* or *Okp1_cmΔ*, with GFP-tagged kinetochore subunits. Image pixels for images of cells with GFP fusion proteins of Chl4, Iml3, Mif2, Ctf3, Cnn1, Nkp1, and Nkp2 (right panel) are 2×2 binned; scale bar: 2 μm. The low GFP signal intensity that we observe for Chl4-GFP and Iml3-GFP in *Okp1_fl* cells is consistent with the low GFP signal for Chl4-GFP and Iml3-GFP at centromeres that was previously reported (Joglekar *et al*, 2008; Lawrimore *et al*, 2011).
E  Scheme summarizing our *in vivo* observed centromere localization dependencies. Solid arrows: localization dependencies found by our study or by studies of others; dashed arrows: protein–protein interactions.

localize to centromeres in *Okp1_cmΔ* (Fig 7D), suggesting that Ctf19-Mcm21 contacts Chl4-Iml3. To test for such an interaction, we combined recombinant samples of Chl4-Iml3 and COMA, from *S. cerevisiae*. Chl4-Iml3 and COMA did not co-elute in size-exclusion chromatography, suggesting that in solution they do not bind each other—in the absence of posttranslational modifications or other macromolecular factors. We found, however, that under similar solution conditions, *in vitro* translated Chl4 associates with COMA (Fig EV6B).

In our microscopy images, centromere GFP signals of Ctf3[CENP-I], Mcm16[CENP-H], Mcm22[CENP-K], Cnn1[CENP-T] or Wip1[CENP-W] in *Okp1_cmΔ* were similar to those in *Okp1_fl* (Fig 7D; Appendix Fig S8D). Similar centromere localization dependencies of these subunits *in vivo* are in agreement with their configuration in a stable assembly *in vitro* (Pekgoz Altunkaya *et al*, 2016). We found, with our experiments, however, that Mcm16 depends on the Ctf19-Mcm21 binding motif for co-immunoprecipitation with Okp1, after isolation from *S. cerevisiae* extracts (Appendix Fig S8E and F).

Nkp1 localized to centromeres in *Okp1_cmΔ*, but substantially less of it than in *Okp1_fl* (Fig 7D). We attribute the reduced localization to disruption of the Okp1 C-terminus in our Okp1_cmΔ variant, essentially in agreement with the—more pronounced—effect in *Okp1_nnΔ* (Fig 5C). The Nkp2-GFP signal was also lower in *Okp1_cmΔ* cells than in *Okp1_fl* cells (Fig 7D), but less so than the Nkp1-GFP signal. This observation is consistent with our biochemical data that indicate separate Ame1-Okp1 binding sites for Nkp1 and Nkp2, and that Nkp1 and Nkp2 bind each other.

We conclude that absence of the Ctf19-Mcm21 binding motif selectively abrogates centromere localization of Chl4, Ctf19, Iml3 and Mcm21; but does not affect centromere localization of subunits of Ctf3-Mcm16-Mcm22, Cnn1-Wip1 or the outer kinetochore (Fig 7E).

### The Ctf19-Mcm21 binding motif is essential in the absence of a functional mitotic checkpoint

Mutant clones *Okp1_cmΔ*, and similar mutant clones that lack coding regions for both the Ctf19-Mcm21 binding motif and the Nkp1-Nkp2 binding motif—*Okp1_cmΔnnΔ* are sensitive to benomyl (a microtubule-depolymerizing chemical) at elevated temperature (Fig 8A; Appendix Fig S9A). On solid standard growth medium, these mutants had similar vegetative growth as *Okp1_fl* (Fig 8A; Appendix Fig S9A), consistent with our observation that the motifs

for binding Ctf19-Mcm21 or Nkp1-Nkp2 are required only for centromere localization of specific non-essential CCAN subunits. To generate more pronounced effects on CCAN structural integrity, we combined *Okp1_cmΔ* (with Ctf19-Mcm21 and Chl4-Iml3 absent from mitotic centromeres) with mutants that lack a specific CCAN subunit, whose mitotic centromere localization in living cells does not depend on the Ctf19-Mcm21 binding motif. For our *Okp1_cmΔ cnn1Δ* mutant clones, we observed reduced growth in the presence of benomyl at 37°C, compared with *Okp1_cmΔ* or *cnn1Δ* (Fig 8A). We observed a similar, but less pronounced, effect for our *Okp1_cmΔ ctf3Δ* mutant clones (Appendix Fig S9B). Our observations suggest that in mitotic cells, absence of multiple CCAN subunits, whose centromere localization does not depend on each other, impairs kinetochore function, but does not abrogate it.

Consistent with similar growth rates, we found that, within a few rounds of cell division in solution, the sister chromatid segregation fidelity of *Okp1_fl* or *Okp1_cmΔ*, both of which we had modified to encode fluorescently labelled chromosome 5, was similar, as judged by the presence of GFP signals from two separated chromatids in large budded cells (Appendix Fig S9C). We noticed, however, that *Okp1_cmΔ* took longer to complete mitosis relative to *Okp1_fl*. The delay was indicated by a higher fraction of large budded *Okp1_cmΔ* cells, which showed kinetochore localization of GFP-tagged mitotic checkpoint protein Bub1 (Appendix Fig S9D), suggesting delay of anaphase onset through an active mitotic checkpoint. To investigate the relevance of the mitotic checkpoint for *Okp1_cmΔ*, we mated *Okp1_cmΔ* or *Okp1_fl* with a mutant that lacks the mitotic checkpoint protein Mad1 (*mad1Δ*). We observed, by tetrad dissection analysis, that haploid *Okp1_cmΔ mad1Δ* spores were either not viable or grew very slowly after germination (Fig 8B). Growth assays of genotyped spores (Fig 8C), or conditional cellular Mad1 degradation (Appendix Fig S9E and F), confirmed our observation. We conclude that in the absence of the Ctf19-Mcm21 binding motif, a functional mitotic checkpoint is essential to maintain viability.

Our analyses show that *Okp1_cmΔ* is benomyl sensitive (Fig 8A) and delays anaphase onset. Its mitotic phenotypes mimic those reported for *chl4Δ, ctf19Δ* or *mcm21Δ* (Hyland *et al*, 1999; Poddar *et al*, 1999; Pot *et al*, 2003), as we expected from our biochemical data and our fluorescence microscopy data. To observe possible effects of the absence of the Ctf19-Mcm21 binding motif on chromosome segregation in meiosis, we evaluated the viability of germinated haploid spores that originated from homozygous diploids of

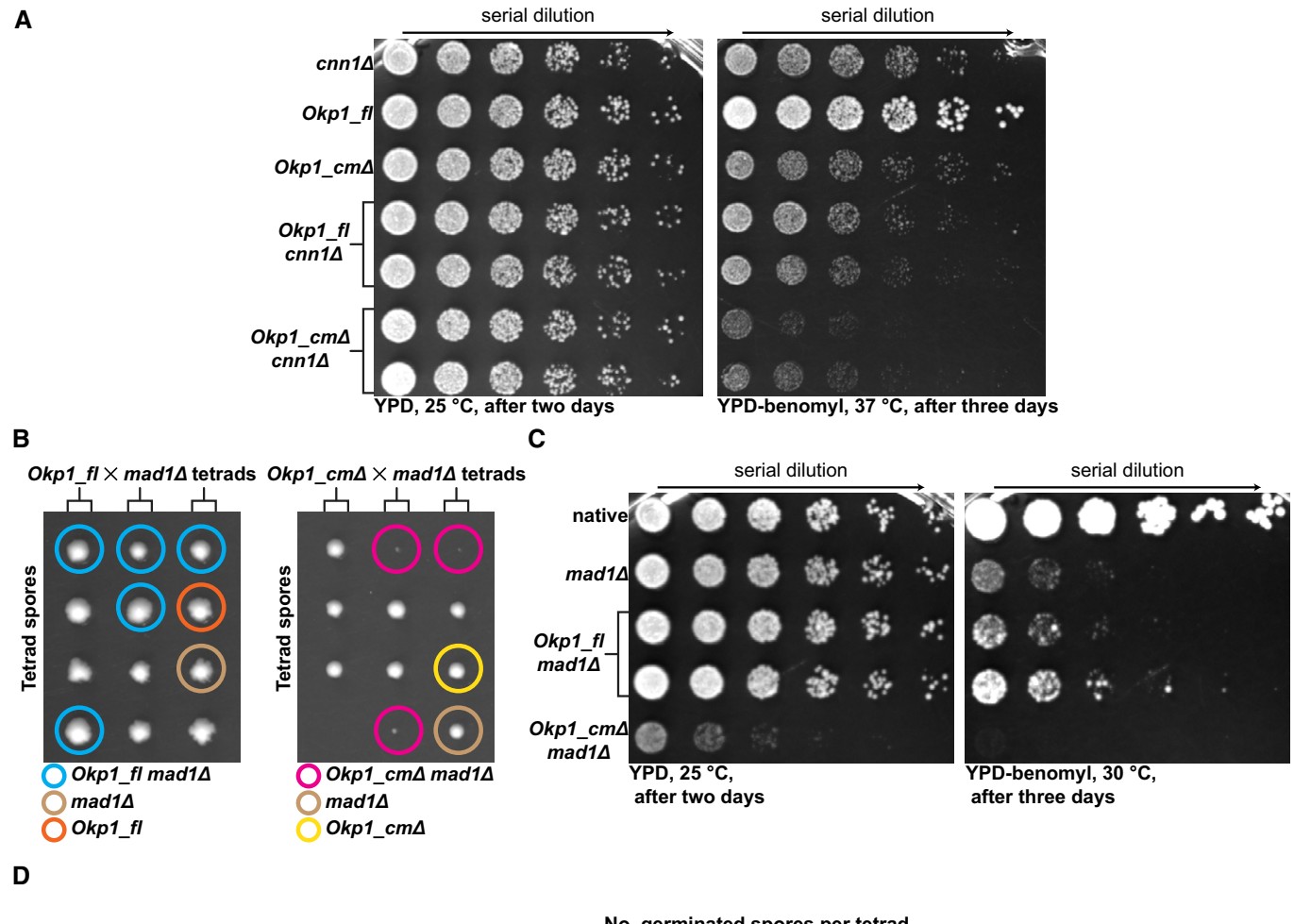

**Figure 8. Relevance of the Ctf19-Mcm21 binding motif for cell viability.**

A Representative images of dilution-series growth assay of haploid *S. cerevisiae* clones with *cnn1Δ*, *Okp1_fl*, *Okp1_cmΔ*, *Okp1_fl cnn1Δ* or *Okp1_cmΔ cnn1Δ*, grown for 2 or 3 days on solid YPD agar or on solid YPD agar with 20 μg/ml benomyl. We show two unique clones each for *Okp1_fl cnn1Δ* or *Okp1_cmΔ cnn1Δ*. Arrow indicates decreasing cell density. Spots are from cell samples that we sequentially diluted fourfold.

B Representative images of haploid *S. cerevisiae* spores from dissection of three tetrads (we placed the four spores from each tetrad row-wise) from heterozygous diploid cells with *Okp1_fl mad1Δ* or *Okp1_cmΔ mad1Δ* grown for 4 and 3 days, respectively, on solid YPD agar at 25°C. Coloured circles indicate spore genotypes (see below the images).

C Representative images of dilution-series growth assay of haploid *S. cerevisiae* with *mad1Δ*, *Okp1_fl mad1Δ* or *Okp1_cmΔ mad1Δ*; native: *S. cerevisiae* S288C type (identifier DDY904 in Table EV6). Dilution is as we describe for (A). We show two unique clones for *Okp1_fl mad1Δ*.

D Quantification of spore viability after meiosis, of spores of dissected tetrads from homozygous diploid *S. cerevisiae* clones with either *Ctf19_1–954Δ* (without *Ctf19* base pairs 1–954; leaving the 3′ end of *Ctf19*, and, on the complementary DNA strand, the *IRC15* gene, which encodes a microtubule binding protein (Keyes & Burke, 2009), intact), *Ctf19Δ* (that lacks the entire *Ctf19*), *Okp1_fl*, *Okp1_cmΔ* or *Okp1_cmΔnnΔ*. Quantification is from spores from multiple tetrad dissections.

*Okp1_fl*, *Okp1_cmΔ* or *Okp1_cmΔnnΔ*, and compared with spore viability of *ctf19Δ* homozygous diploids. Viability of spores from *ctf19Δ/ctf19Δ*, of *S. cerevisiae* SK1 laboratory type, was previously reported to be low (Mehta *et al*, 2014). We found that spore viability from *ctf19Δ/ctf19Δ*, of our type of *S. cerevisiae* (S288C), was

indeed low (37% for *ctf19_1–954Δ/ctf19_1–954Δ;* Fig 8D; Appendix Fig S9G), suggesting pronounced chromosome mis-segregation during meiosis. For *Okp1_cmΔ/Okp1_cmΔ*, spore viability, unexpectedly, was only moderately decreased (86%), compared with *Okp1_fl/Okp1_fl* (97%) (Fig 8D; Appendix Fig S9G). We

conclude that absence of the Ctf19-Mcm21 binding motif affects meiotic kinetochores and mitotic kinetochores differentially.

## Discussion

Although centromere-associated proteins evolve rapidly, in animals and plants presumably due to "centromere drive" (Henikoff *et al*, 2001; Drinnenberg *et al*, 2014), budding yeasts' kinetochores, which assemble on "point" centromeres, are surprisingly similar to human kinetochores and fission yeasts' kinetochores, which assemble on "regional" centromeres, in protein composition and structural features (Westermann & Schleiffer, 2013). The orthologues for Ame1 and Okp1 (Schleiffer *et al*, 2012), which are essential for *S. cerevisiae*, are in vertebrates CENP-U/CENP-50 (Minoshima *et al*, 2005) and CENP-Q (Okada *et al*, 2006), respectively, and in *Schizosaccharomyces pombe* Mis17 (Hayashi *et al*, 2004) and Fta7 (Shiroiwa *et al*, 2011), respectively. CENP-U and CENP-Q are, however, not essential for chicken cell lines (Minoshima *et al*, 2005; Okada *et al*, 2006; Hori *et al*, 2008), although CENP-U-deficient mouse embryos died (Kagawa *et al*, 2014). Yet, little mechanistic data have been reported for the function of CENP-O/P/Q/U or COMA.

### Okp1 is a multi-segmented kinetochore nexus

Our biochemical and structural analyses show the contributions of Okp1 to inner kinetochore organization. We defined in Okp1 three segments, which are spatially separated by flexible elements, that have distinct binding sites for different inner kinetochore proteins (Fig 9A). We suggest that Okp1 is a multi-segmented molecular nexus.

We characterized the Ctf19-Mcm21 binding motif as the principal contact of Okp1 with Ctf19[CENP-P]-Mcm21[CENP-O] (Fig 9A; Movie EV1). We previously showed that the *K. lactis* Ctf19-Mcm21 D-RWD domains, which are probably structurally similar in humans, suffice to associate Ctf19-Mcm21 with Okp1-Ame1 (Schmitzberger & Harrison, 2012). We have now shown that Okp1 binds the hydrophobic Ctf19 RWD-C surface, and central helices in the Mcm21 D-RWD domain. Our structure of Ctf19-Mcm21 with the Ctf19-Mcm21 binding motif is the first described example of an RWD domain–peptide assembly from the inner kinetochore. We did not find major structural similarities in RWD domain–peptide interactions between our structure and those of the other reported kinetochore RWD domains bound with peptides—Csm1 with a Mam1 peptide (Corbett & Harrison, 2012), Knl1 with an Nsl1 peptide (Petrovic *et al*, 2014) and Spc24-Spc25 either with a Cnn1[CENP-T] peptide (Malvezzi *et al*, 2013; Nishino *et al*, 2013) or with a Dsn1 peptide (Dimitrova *et al*, 2016) (Fig 9B). We conclude that binding modes of peptides to kinetochore RWD domains differ. Distinct modes ensure specificity in peptide recognition by RWD domains in kinetochore assembly. Our deuterium-exchange analyses show that, when bound to each other, Ctf19-Mcm21 RWD domains and Ctf19-Mcm21 binding motif of Okp1 stabilize each other, which probably contributes to binding specificity and kinetochore stability. Apart from a few residues that make important contacts in our structure, the Ctf19-Mcm21 binding motif's sequence is, however, not very similar among budding yeasts (Fig EV5). Likewise, the residues of

Ctf19 or Mcm21 that contact Okp1 are not conserved among budding yeasts. We explain the low similarity or absence of sequence conservation by co-evolution of the binding sites in both Ctf19-Mcm21 and Okp1, and main chain contacts that are phylogenetically less restrained by residue identity. We were unable to identify, by sequence comparison, a corresponding CENP-P/O binding motif in CENP-Q sequences (Fig EV7). We note, however, that a short α-helix, possibly related to the Ctf19-Mcm21 binding motif, is predicted in CENP-Q sequences for the region that corresponds to the motif in Okp1 sequences.

Our data suggest that Okp1 segment 2 binds Ame1 core, probably through a coiled coil (Fig 9A and C). This suggestion is supported by cross-linking data, which show that several lysines in the C-terminal part of Ame1 core are proximal to those C-terminal of Okp1 segment 2 (Hornung *et al*, 2014). Our *S. cerevisiae* mutant that lacks Okp1 segment 2 is not viable (Appendix Fig S3D), consistent with our suggestion that segment 2 interacts with parts of another essential kinetochore subunit (Ame1). Sequences of Ame1 core and Okp1 segment 2 are very similar among budding yeasts' Ame1 proteins and Okp1 proteins, respectively (Fig EV5; Appendix Fig S3F). The equivalent segments in CENP-U proteins (Appendix Fig S10) and CENP-Q proteins (Fig EV7) probably are coiled coils too, suggesting that an Ame1-Okp1 coiled coil, like the joint Ctf19-Mcm21 D-RWD modules, is a structural feature conserved between yeasts and humans. It is thus plausible that a similar separation of binding sites for Ame1[CENP-U] or Ctf19[CENP-P]-Mcm21[CENP-O] that we found in Okp1[CENP-Q] is present in CENP-Q proteins.

Our data show that Okp1 segment 3, with probable contribution from Ame1 segment 1, binds Nkp1-Nkp2. In our nanoflow mass spectra, we found a signal that corresponds to Nkp1-Okp1 (-Ctf19-Mcm21; Fig EV2B), suggesting that Nkp1 is the primary Okp1 binding partner of Nkp1-Nkp2. This suggestion is consistent with our observation in living cells (of *Okp1_nnΔ*) that Nkp1 and Nkp2 barely localize to kinetochores in the absence of Okp1 segment 3 (Fig 5C). We presume that Ame1 segment 1 and Okp1 segment 3 bind Nkp2 and Nkp1, respectively, which can explain why centromere localization of Nkp2 in cells without Okp1 segment 3 was less abrogated than that of Nkp1. Nkp1 and Nkp2 presumably exist mainly as a heterodimer in living cells. Because Nkp1-Nkp2 presumably brings the Ame1 C-terminus and the Okp1 C-terminus in proximity of one another, there could be a composite binding site. The C-termini of Okp1 or Ame1 are, however, not conserved in sequence. We found that Ame1 and Okp1 from *Eremothecium gossypii*, and from a few other budding yeasts that include *Eremothecium cymbalariae* and *Naumovozyma dairenensis*, entirely lack C-terminal segments that are equivalent to those of *K. lactis* Ame1 segment 1 or the Nkp1-Nkp2 binding site in *K. lactis* Okp1, respectively (Fig EV5; Appendix Fig S3F). Corresponding to this observation, for *E. gossypii*, *E. cymbalariae* and *N. dairenensis*, we were unable to identify orthologues of Nkp1 or Nkp2. We conclude that the Nkp1-Nkp2 function is directly linked with the Okp1 C-terminus and the Ame1 C-terminus. Our data show that Nkp1-Nkp2 binding makes these termini less flexible. We reason that Nkp1 and Nkp2 have regulatory roles for C-termini of Ame1 and Okp1, which are auxiliary to inner kinetochore stability. With human CENP-O/P/Q/U co-purifies CENP-R (Okada *et al*, 2006). Although not similar in sequence to Nkp1 or Nkp2, it may be their functional counterpart.

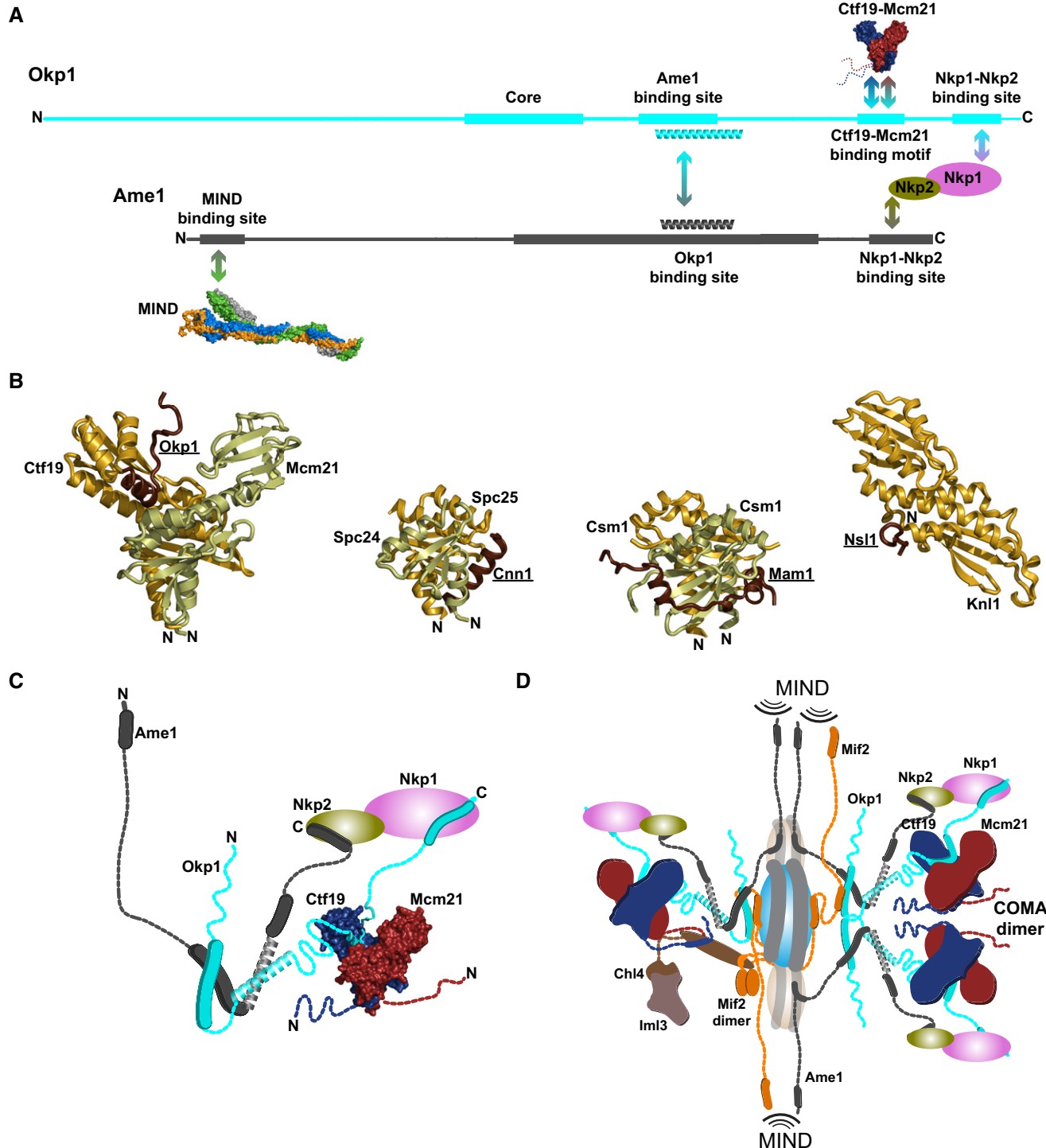

**Figure 9.  Molecular interactions of Okp1, Ame1 and RWD domains in kinetochores.**

A   Scheme of binding sites of Okp1 and Ame1 in budding yeasts' kinetochores that we identified; we also show the previously identified binding site for MIND (PDB code: 5T58) in Ame1.

B   Secondary structure cartoon diagrams of kinetochore RWD domain–peptide structures. Ctf19-Mcm21 with Okp1 peptide (our study); *S. cerevisiae* Spc24-Spc25 with Cnn1 peptide (PDB code: 4GEQ); *S. cerevisiae* Csm1 with Mam1 peptide (5KTB; we show Csm1 residues 67–181); and human Knl1 with Nsl1 peptide (4NF9). We show structures in orientations superposed on Ctf19 and on the same scale.

C   Schematic illustrating binding sites and dynamic regions in COMA-Nkp1-Nkp2.

D   Schematic of a partial cross section of budding yeasts' inner kinetochore around a centromeric nucleosome (blue, viewed from the outer kinetochore). Right side illustrates a dimeric COMA assembly; left side, binding of Chl4-Iml3 to Ctf19-Mcm21 and Mif2. We show identified contacts and a few presumed contacts. Interaction sites with the MIND assembly, which connects the inner kinetochore to the outer kinetochore, are indicated.

The largest structured Okp1 segment—Okp1 core (Fig 9A)—is a possible contact site for Mif2$^{CENP-C}$, since Okp1-Ame1 binds Mif2. Several lysines in the conserved region that is preceding Okp1 core cross-linked with lysines in the Mif2 "signature" sequence (Hornung *et al*, 2014), suggesting these parts are proximal to each other. We have shown that this Mif2 sequence is required for binding COMA.

A small N-terminal motif in Ame1 (Appendix Fig S3F), which does not seem to be present in human CENP-U (Appendix Fig S10), is essential for *S. cerevisiae* for outer kinetochore assembly by binding MIND (Hornung *et al*, 2014; Dimitrova *et al*, 2016; Fig 9A). In the absence of MIND, our deuterium-exchange data show that this motif is disordered (Fig 3B). We conclude that Ame1, like Okp1, has multiple contact sites for kinetochore proteins, which are spatially separated by flexible elements. Okp1, Ame1 and Mif2 (Cohen *et al*, 2008), which are essential for *S. cerevisiae* and associate with each other, are a group of dynamic multi-segmented molecular nexuses at the inner kinetochore. The elasticity of their flexible elements is presumably important during the dynamic events of kinetochore–microtubule attachment and chromosome movement. We conclude that inner kinetochore molecular organization is defined by flexible molecular nexuses and globular RWD domains and small peptides that form very stable interactions with each other (Fig 9D).

Except for the Ame1 binding site and Okp1 core, Okp1 sequence features differ from those of orthologous CENP-Q and Fta7 (Fig EV7), which have fewer residues than Okp1. Most of the additional residues in Okp1 are in the sequence feature-variant, flexible N-terminus. Except for the conserved Ame1 core, Ame1 also differs substantially in sequence from CENP-U and Mis17, which have more residues than Ame1 (Appendix Fig S10). Most of the additional residues are N-terminal of Ame1 core. The N-terminal regions of Ame1, Okp1, CENP-U (Hori *et al*, 2008) and Mis17 (Shiroiwa *et al*, 2011) are phosphorylated and may, as proposed for Mis17, primarily have regulatory roles. We conclude from our sequence comparisons that COMA and CENP-O/P/Q/U have structural features in common, although overall sequence features diverged.

## Molecular assembly of COMA at kinetochores

We have shown that Ame1 does not contact Ctf19-Mcm21 and that its centromere localization is unaffected by the absence of Ctf19-Mcm21 and Chl4-Iml3 from mitotic centromeres. Previous analyses of temperature-sensitive *S. cerevisiae* clones with mutants of *Ame1* or *Okp1* indicated that for centromere localization, Ctf19, Mcm21 and Okp1 depended on Ame1, but Ame1 did not depend on the other COMA subunits (Pot *et al*, 2005). These data suggest that Ame1 is required, through its interaction with Okp1, to localize Ctf19-Mcm21-Okp1 to mitotic centromeres.

We have shown that COMA tends to dimerize through Ame1-Okp1. At kinetochores, a dimer is presumably stabilized by contacts that increase COMA's effective concentration (Fig 9D). Consistent with this presumption, we have shown that COMA-MIND dimerizes (Fig EV1A). Measurements in human cells indicated that CENP-O/P/Q/U oligomerizes through CENP-Q/U at mitotic kinetochores (Eskat *et al*, 2012). Quantification from live cell fluorescence microscopy yielded estimates that are consistent with three to eight Ctf19 molecules and up to four Mif2 molecules

for each *S. cerevisiae* kinetochore (Joglekar *et al*, 2006; Lawrimore *et al*, 2011). CENP-C$^{Mif2}$ binds CENP-A$^{Cse4}$ nucleosomes (Kato *et al*, 2013), and Mif2 dimerizes (Cohen *et al*, 2008). Our data suggest that COMA multimerization is an important effector for amplification of the Cse4 kinetochore nucleation signal, through provision of multiple binding sites to MIND, for outer kinetochore assembly (Fig 9D).

## The Ctf19-Mcm21 binding motif configures a branch of functionally related inner kinetochore subunits

Our characterization of COMA guided us to analyse the specific role of the Ctf19-Mcm21 binding motif for kinetochore organization in living cells. Mutant *Okp1_cmΔ*, without the Ctf19-Mcm21 binding motif, is defective in localizing Ctf19$^{CENP-P}$-Mcm21$^{CENP-O}$ and Chl4$^{CENP-N}$-Iml3$^{CENP-L}$ to mitotic centromeres. Our observations are in agreement with the previously described centromere localization dependence of Chl4-Iml3 on Ctf19 in diploid *S. cerevisiae* cells (Pot *et al*, 2003). Chl4 binds Mif2$^{CENP-C}$ *in vitro* (Hinshaw & Harrison, 2013), but—in the absence of Ctf19-Mcm21—this interaction does not suffice to centromere-localize Chl4-Iml3 *in vivo*. Because we did not observe association of Chl4-Iml3 with COMA in solution, their stable association presumably requires simultaneous binding by Mif2. Our binding experiments (Fig EV6) suggest that Chl4-Iml3 interacts through Chl4 with Ctf19-Mcm21. An interaction of Chl4 may be with the parts N-terminal of the Ctf19-Mcm21 D-RWD domains, which contain conserved residues and were disordered in crystals of full-length Ctf19-Mcm21 (Schmitzberger & Harrison, 2012). The D-RWD domains themselves have few conserved surface residues outside of the Okp1 binding site.

Our live cell microscopy images show that mitotic centromere localization of Ctf3$^{CENP-I}$, Mcm16$^{CENP-H}$, Mcm22$^{CENP-K}$, Cnn1$^{CENP-T}$ or Wip1$^{CENP-W}$, which form an assembly that contacts the NDC80 assembly (Pekgoz Altunkaya *et al*, 2016), does not depend on the Ctf19-Mcm21 binding motif, and—by extension—on Ctf19-Mcm21 and Chl4-Iml3. Previous reports described that Ctf3 localized to centromeres in living anaphase mutant cells that lacked *Ctf19* or *Chl4* (Pot *et al*, 2003). In immunoprecipitated isolates from *S. cerevisiae* cell extracts, however, Ctf3 depended on Ctf19 or Mcm21 for co-immunoprecipitation with Ame1 or centromeric DNA (Measday *et al*, 2002; Pekgoz Altunkaya *et al*, 2016), and Mcm16 depended on the Ctf19-Mcm21 binding motif for co-immunoprecipitation with Okp1 (Appendix Fig S8E and F), suggesting that conditions in extracts do not fully reflect native-like centromere localization requirements for Ctf3 or Mcm16. We assume that chromatin-binding motifs, such as present in Cnn1 and Wip1, and DNA-binding activities, which remain to be characterized, contribute to localization of Ctf3 or Mcm16 to centromeres in living cells.

We conclude that the Ctf19-Mcm21 binding motif in Okp1 configures a "branch" of functionally related subunits of the CCAN assembly, by tethering Ctf19-Mcm21 and—indirectly—Chl4-Iml3 to mitotic centromeres (Fig 9D). This motif defines a kinetochore assembly axis that is parallel to the assembly axis for the outer kinetochore, which is based on contacts of Ame1 or Mif2 with Mtw1-Nnf1, and Cnn1 or Dsn1 with Spc24-Spc25.

In contrast to the effect of absence of Ctf19-Mcm21 on Chl4-Iml3 centromere localization in *S. cerevisiae*, absence of CENP-O$^{Mcm21}$ or

CENP-P$^{Ctf19}$ did not affect CENP-L$^{Iml3}$ centromere localization in human cells (Okada *et al*, 2006; McKinley *et al*, 2015), nor recombinant human CENP-N$^{Chl4}$-CENP-L$^{Iml3}$ binding to reconstituted CENP-A$^{Cse4}$ nucleosomes (Weir *et al*, 2016). We conclude that between budding yeasts' COMA or Chl4-Iml3, and their mammalian orthologues CENP-O/P/Q/U or CENP-N/L, respectively, there are important differences in contacts with other mitotic kinetochore subunits. Protein connections in CCAN evolved.

### Relevance of the Ctf19-Mcm21 binding motif for chromosome segregation

The pronounced dependence of cells that lack the Ctf19-Mcm21 binding motif on the mitotic checkpoint suggests that, in the absence of Ctf19-Mcm21 and Chl4-Iml3, these cells have kinetochore–microtubule attachment errors. If cell cycle progression is not delayed by the mitotic checkpoint for correction of these errors, these errors result in chromosome mis-segregation. Mis-segregation rates of artificial chromosomes or native chromosomes were indeed elevated in mitotic cells with *chl4Δ*, *ctf19Δ* or *mcm21Δ* (Hyland *et al*, 1999; Poddar *et al*, 1999; Pot *et al*, 2003; Fernius & Marston, 2009), and a relevance of Ctf19$^{CENP-P}$ for mitotic checkpoint function was reported (Matson *et al*, 2012). We conclude that mitotic kinetochore function is impaired in the absence of the Ctf19-Mcm21 binding motif.

One probable source of kinetochore–microtubule attachment errors in mitotic cells without the Ctf19-Mcm21 binding motif is a pericentromeric cohesion defect. Pericentromeric cohesin loading in budding yeasts' meiosis and mitosis depends on Ctf19-Mcm21 and Chl4-Iml3 (Fernius & Marston, 2009; Natsume *et al*, 2013). Pericentromeric cohesion facilitates mitotic kinetochore biorientation (Ng *et al*, 2009) and meiotic sister chromatid co-orientation. Absence of Chl4, Ctf19, Iml3 or Mcm21 in meiosis results in chromosome/chromatid non-disjunction and aneuploidy (Fernius & Marston, 2009; Mehta *et al*, 2014). In contrast to the pronounced meiotic phenotype of homozygous diploid *Ctf19* deletion mutants, the phenotype of our homozygous diploid *Okp1_cmΔ* mutant, which we had expected to be similar to that of *Ctf19Δ*, was near native. Our finding suggests that, instead of the Ctf19-Mcm21 binding motif, other factors retain Ctf19-Mcm21 at meiotic kinetochores. We conclude that there are important structural differences between inner meiotic kinetochores and inner mitotic kinetochores. Recent studies reported differences between outer meiotic kinetochores and outer mitotic kinetochores (Mehta *et al*, 2014; Meyer *et al*, 2015). Exploring inner kinetochore structural differences, and identifying the factors accounting for them, will be relevant to understand kinetochore plasticity.

Our presented data inform on structure, topology and subunit connections of the inner mitotic kinetochore, and contribute to a conceptual framework for further characterization of its dynamic architecture.

## Materials and Methods

For a detailed description of our materials and methods, see our Appendix Supplementary Materials and Methods section.

### Molecular cloning, site-directed mutagenesis, recombinant protein production and protein purification

We constructed most of our polycistronic plasmids similarly as previously described (Schmitzberger & Harrison, 2012), by integrating coding regions (for full-length proteins or truncation variants) into a pET-based plasmid (with T7 DNA polymerase promoter) encoding a tobacco-etch virus (TEV) protease cleavable N-terminal polyhistidine tag. Coding regions for *S. cerevisiae* COMA were inserted into a pST39 plasmid (Tan, 2001). We generated our *K. lactis* COMA variant that lacks the coding region for Okp1 segment 1 by QuikChange-based site-directed mutagenesis with *PfuTurbo* DNA Polymerase AD (Agilent Technologies). For a list of our plasmids and constructs, see Table EV5. Plasmids are available upon request. We produced recombinant proteins in *Escherichia coli* BL21 Rosetta 2(DE3)pLysS cells, usually in terrific broth, by induction with isopropyl-β-D-thiogalactopyranoside (IPTG). We adapted our small-scale 96-well plate-based screening method for protein production and Ni$^{2+}$ affinity chromatography protein purification from previously described methods (Savitsky *et al*, 2010). For large-scale purification, we purified proteins first with Ni$^{2+}$ affinity chromatography at 4°C. Most of our proteins we subsequently incubated with TEV protease (Kapust *et al*, 2001) for 16–18 h at 4°C. After another Ni$^{2+}$ affinity purification step, we purified proteins from the affinity chromatography flow-through fraction with ion-exchange chromatography (5 ml HiTrap Q HP column or 5 ml HiTrap SP HP column, GE Healthcare Life Sciences), followed by SEC (Superdex 200 HiLoad 16/600 prep grade column; GE Healthcare Life Sciences) at 4°C. We produced and purified *S. cerevisiae* Okp1-Ame1 similarly as previously described (Hornung *et al*, 2014).

### Hydrogen-deuterium exchange coupled to mass spectrometry

For our comparative deuterium-exchange experiments, we used purified protein samples that we had flash-frozen in liquid $N_2$ and had stored at −80°C and on solid $CO_2$. We carried out deuterium-exchange experiments similarly as previously described (Kupniewska-Kozak *et al*, 2010), with a reaction buffer containing $D_2O$ (99.8% (v/v); Armar Chemicals). We quenched deuterium-exchange reactions by reducing the pH to ~2.5 with 2 M glycine. We digested deuterium-exchanged proteins using an immobilized pepsin column (Poroszyme, Applied Biosystems, Thermo Fisher Scientific). We injected digested peptides into the nanoACQUITY (Waters Corporation) Ultra Performance Liquid Chromatography (UPLC) system, and separated peptides over a C18 trapping column (ACQUITY BEH C18 VanGuard Pre-column, Waters Corporation), followed by separation over a reversed-phase chromatography column (ACQUITY UPLC BEH C18 column, Waters Corporation). The outlet of the latter column was coupled directly to the ion source of a SYNAPT G2 HDMS mass spectrometer (Waters Corporation). We identified peptides with the ProteinLynx Global Server software (Waters Corporation). We carried out two kinds of control experiments to determine experimental in-exchange—our minimum exchange value ($M_{ex}^0$), or back-exchange values—our maximum exchange value ($M_{ex}^{100}$), as previously described (Kupniewska-Kozak *et al*, 2010). We calculated the deuteration level of peptides ($M_{ex}$) with DynamX (Waters Corporation), using as reference the m/z values from non-deuterated pepsin-proteolysed peptides that we

calculated with the ProteinLynx Global Server. We calculated the fraction of peptide deuterium exchange (*f*) with the formula:

$$f = \frac{M_{ex} - M_{ex}^0}{M_{ex}^{100} - M_{ex}^0}$$

We calculated mean value and standard deviation for *f* from at least three independent experiments. We visualized our exchange data with a previously described Excel macro (Black *et al*, 2007) that plots deuterium-exchanged peptide representations that are coloured by fraction of deuterium exchange, underneath their corresponding position in a linear amino acid sequence representation of the protein the respective peptides originate from. Our raw deuterium-exchange data are available upon request.

### Nanoflow electrospray ionization mass spectrometry

For our nanoflow electrospray ionization mass spectrometry, we used purified protein samples that we had flash-frozen in liquid $N_2$ and had stored at −80°C and on solid $CO_2$. We transferred proteins to a buffer of 200 mM ammonium acetate pH 6.7–7.3 or pH 7.4, and sprayed protein samples, usually at a concentration of 2–10 μM, with in-house prepared gold-coated glass capillaries (Nettleton *et al*, 1998). We acquired mass spectra or tandem mass spectra (Benesch *et al*, 2006), in positive ion mode, on a high mass Q-TOF-type instrument (Sobott *et al*, 2002) adapted for a QSTAR XL platform (MDS Sciex) (Chernushevich & Thomson, 2004). For collision-induced dissociation, we used argon as a collision gas at maximum pressure. We carried out partial, in-solution disruption of purified *K. lactis* COMA (Fig 2A and B; Appendix Fig S2A), by adding acetic acid to a concentration of 5% (v/v) in 100 mM ammonium acetate, to a pH of 4.0. For our analysis of Ctf19-Mcm21-Okp1 that we show in Appendix Fig S2B, we analogously incubated COMA with 100 mM ammonium acetate pH 3.7, 5% (v/v) acetic acid.

### Limited proteolysis followed by mass spectrometry

We used purified protein samples, which we had stored on ice after purification, for our limited proteolysis experiments with trypsin (Sigma-Aldrich) or elastase (Worthington Biochemical Corporation). We incubated COMA-Nkp1-Nkp2 in a 10:1 molar ratio with trypsin for 180, 240 s or 600 s, or in a 10:1 molar ratio with elastase for 180 s, at 22–25°C. We incubated Ame1-Ctf19$_{D-RWD}$-Mcm21$_{D-RWD}$-Okp1 in a 10:1 molar ratio with trypsin for 255 s, at 22–25°C. We stopped our proteolysis reactions by adding 4-(2-aminoethyl)benzenesulfonyl fluoride hydrochloride (AEBSF) solution to a concentration of ~1.5 mM, and gel-filtered samples on a Superdex 200 HiLoad 16/600 prep grade column (Fig EV4). We pooled fractions corresponding to the principal elution peak, and denatured samples by the addition of solid high-grade guanidine hydrochloride (cat. no 50933, Sigma-Aldrich) until saturation of the solution. For limited proteolysis without subsequent SEC, we incubated COMA-Nkp1-Nkp2 in a molar ratio of 10:1 with elastase or trypsin, for 180 s or 600 s, at 22–25°C; we incubated *K. lactis* Nkp1-Nkp2 in a molar ratio of 10:1 with elastase or trypsin, for 300 s or 600 s, at 22–25°C, before the addition of solid high-grade guanidine hydrochloride.

David S. King analysed protein fragments with Fourier transform ion resonance cyclotron or ion-trap mass spectrometers at the Howard Hughes Medical Institute mass spectrometry facility (University of California, Berkeley), and analysed mass spectra. For information about our protein fragments that we identified, see Tables EV1–EV3.

### Binding assays with *in vitro* translated proteins

Plasmids or PCR products with coding regions for proteins for *in vitro* translation with a Kozak translation initiation sequence we prepared as previously described (Hinshaw & Harrison, 2013). We produced $S^{35}$-labelled proteins by *in vitro* translation in rabbit-reticulocyte lysate (TnT lysate systems; Promega Corporation) with $S^{35}$ L-methionine, following the manufacturer's instructions. For our binding assays, we used *K. lactis* COMA that we had stored on ice or at −80°C after purification, or *S. cerevisiae* COMA and poly-histidine-tagged maltose binding protein that we had stored at −80°C.

### Analytical size-exclusion chromatography, multi-angle laser light scattering measurements and dynamic light scattering measurements

We carried out analytical SEC for samples that we show chromatograms in Figs 5A and B, and EV3A on a Superdex 200 10/300 column (GE Healthcare Life Sciences) on an Äkta FPLC (GE Healthcare Life Sciences) system, at 4°C. For our sample analysis that we show chromatograms in Fig EV3B, we carried out SEC with a Superdex 200 PC 3.2/300 column (GE Healthcare Life Sciences) on an Ettan LC system (GE Healthcare Life Sciences). For our analysis that we show chromatograms in Appendix Fig S7B, we used a Superdex 200 HiLoad 10/600 prep grade column. For details, see our Appendix Supplementary Materials and Methods. We measured multi-angle laser light scattering data on a Dawn Heleos-II detector (Wyatt-846-H2; Wyatt Technology) and refractive indices with an Optilab T-rEX instrument (Wyatt-512-Trex; Wyatt Technology), eluting proteins from a Superdex 200 10/300 SEC column that was mounted on a high-performance liquid chromatography system (1260 Infinity LC; Agilent Technologies). We analysed multi-angle laser light scattering data with the Astra software (Wyatt Technology), with protein concentrations determined from in-line refractive index measurements and using a dn/dc value of 0.185 ml/g (a refractive index of 1.33 was chosen for the aqueous solution). We fit our multi-angle laser light scattering data with a first-order Zimm function with linear regression, as implemented in the Astra software. We measured dynamic light scattering data of *K. lactis* COMA in a quartz cuvette on a Dynapro instrument (Wyatt Technology) with a laser of 826.2 nm, at 15°C. We fit monomodal auto-correlation functions for our data with the Dynamics software (Wyatt Technology). Our light scattering data are available upon request.

### Sedimentation-equilibrium analytical ultracentrifugation analyses

For our sedimentation-equilibrium analytical ultracentrifugation, we used purified protein samples, which we had stored on ice after

purification. We recorded sedimentation-equilibrium analytical ultracentrifugation data with a ProteomeLab Optima XL-I analytical ultracentrifuge (Beckman Coulter) and an An-60 Ti rotor (Beckman Coulter) equipped with a 12-mm-wide Epon six-chamber double-sector sample cell. We calculated estimates of the partial specific protein assembly volumes (based on the molecular weight expected from the protein sequences), buffer density and buffer viscosity with SEDNTERP (http://www.jphilo.mailway.com/download.htm). We fit our data with SEDPHAT (http://www.analyticalultracentrifugation.com/sedphat/sedphat.htm). We visualized our data with GUSSI (http://biophysics.swmed.edu/MBR/software.html). For details about our fitting procedure, see our Appendix Supplementary Materials and Methods.

### Isothermal calorimetry titration (ITC) measurements and microscale thermophoresis (MST) measurements

For our ITC or MST measurements, we used Okp1-derived peptides synthesized by Mathias Madalinski at the Protein Chemistry core facility (IMP, Vienna). We carried out ITC measurements on a MicroCal VP-ITC calorimeter (GE Healthcare Life Sciences). We transferred 1.42 ml of Ctf19-Mcm21 sample solution at 13 μM to the calorimeter reaction cell that was kept at 25°C. Okp1-derived peptide, with the Ctf19-Mcm21 binding motif, at 120 μM was injected into the reaction cell; the first injection was with 5 μl (over 10 s), followed by 29 injections, each lasting 20 s, of 10 μl. Injections were repeated every 300 s. The ITC reaction cell was under continuous stirring at 307 rpm. We subtracted from data of these measurements the reference signal of our Okp1-derived peptide titrated to ITC/MST buffer. We used Origin software (OriginLab) to derive the $K_d$, by fitting a non-linear single set of sites binding function with chi$^2$ minimization to our data. We did three independent measurements that yielded similar ITC data, from which we derived mean value and standard error of regression (Fig 6F). For our MST measurements (Wienken *et al*, 2010), we labelled purified Ctf19-Mcm21 with a red fluorescent dye (NT-647-NHS), which covalently modifies lysine side chains, in company-provided "labelling buffer" at 22–25°C, according to the manufacturer's instructions (Nanotemper Technologies), with a 1:4 molar ratio of Ctf19-Mcm21 and dye. We transferred fluorescently labelled Ctf19-Mcm21 and Okp1-derived peptide, or variants of Okp1-derived peptide (Fig 6G), to ITC/MST buffer. We prepared dilution series of peptides, and combined with a uniform volume of Ctf19-Mcm21 in PCR tubes, before transferring to NT.115 MST premium-coated glass capillaries (Nanotemper Technologies). For all our measurements, the total Ctf19-Mcm21 concentration in the capillaries was ~27 nM. We measured MST data on an NT.115 Monolith BLUE/RED instrument (Nanotemper Technologies) at 22–25°C. Samples were heated with an infrared laser (λ: 1,474 nm ± 15 nm) set to 20% MST power (30 s on; 5 s off), and fluorophores excited with a laser (excitation λ: 625 nm; emission λ: 680 nm) set to 60% LED power. For all our MST measurements, fluorescence intensity counts were 350–500. We derived our binding data from the "thermophoresis + T jump" fluorescence signal. For each unique peptide sample, we measured data from three separately prepared dilution series. For our MST data analysis, we used NTAffinityAnalysis software

(version 2.0.2; Nanotemper Technologies). We derived $K_d$ values by least-squares fitting the following law of mass action function, as previously described (Seidel *et al*, 2013):

$$y = U + (B - U) * \frac{(x + c_{labelled} + K_d - \sqrt{(x + c_{labelled} + K_d)^2 - 4 * x * c_{labelled}})}{2 * c_{labelled}}$$

$U$: fluorescence of unbound Ctf19-Mcm21 (base level); $B$: fluorescence of bound Ctf19-Mcm21 (saturation level); $c_{labelled}$: concentration of labelled protein; $K_d$: dissociation constant; $x$: concentration of peptide; $y$: fluorescence.

### Electron microscopy

For electron microscopy (Appendix Fig S2D), our final purification step for *K. lactis* COMA was with SEC with a buffer of 25 mM HEPES pH 7.5, 200 mM NaCl and 0.5 mM TCEP. We stored purified protein samples on ice for ca. 16–24 h after the final purification step, before transferring them, at a concentration of ~70 nM, onto carbon-coated copper-mesh grids (CF-400-Cu, Electron Microscopy Science). We stained COMA samples with uranyl formate. We imaged with a T12 (FEI Tecnai) transmission electron microscope, usually at 80 kV accelerating voltage.

### Protein crystallization, crystal structure determination and coordinate refinement

We crystallized our minimized *K. lactis* Ctf19$_{107–270}$-Mcm21$_{108–293}$-Okp1$_{295–360}$ protein assembly, which we had stored on ice after purification, with 35% (v/v) glycerol ethoxylate and 200 mM Li-citrate. Crystals with dimensions of 50 μm × 50 μm × 50 μm – 100 μm × 100 μm × 100 μm grew typically within 1–4 days, at 20°C. From our crystals, cryo-cooled at −173.5°C, we collected X-ray diffraction data at beamline 24-ID-E of the Advanced Photon Source (Argonne National laboratory) with a charge-coupled device detector (Quantum 315; Area Detector Systems Corporation) and a microdiffractometer. We indexed and integrated X-ray diffraction data with XDS (Kabsch, 1993), and scaled our integrated data with Aimless (Evans & Murshudov, 2013) of the CCP4 suite (Winn *et al*, 2011), keeping Friedel pairs separated. We determined our crystal structure in space group P22$_1$2$_1$, by molecular replacement with the coordinates of the Ctf19-Mcm21 (PDB code: 3ZXU; Schmitzberger & Harrison, 2012) D-RWD domains (Ctf19$_{107–270}$-Mcm21$_{108–293}$) with phenix Phaser (McCoy *et al*, 2007). We used phenix.AutoBuild (Adams *et al*, 2010) for initial rebuilding and refinement of our structure. We subsequently refined our model with phenix.refine (Afonine *et al*, 2012), with a maximum-likelihood target function with a test set of 2.4% of randomly selected reflection indices, and manually rebuilt our model in σ$_A$-weighted electron density maps with COOT (Emsley & Cowtan, 2004), until convergence of the $R_{free}$. We used crystallographic data analyses software provided by SBGrid (Morin *et al*, 2013). We prepared our molecular structure representations (Figs 6 and 9; Appendix Fig S7C and E) and movie (Movie EV1) with PyMOL Molecular Graphics System (Schrödinger, LLC). For details of data collection, coordinate refinement and final model quality, see Table EV4.

## Analysis of protein structure and amino acid sequences

We evaluated our protein model coordinates with Molprobity (Chen *et al*, 2010) and WHATCHECK (Hooft *et al*, 1996). For pairwise superposition of structure coordinates, we used LSQKAB (Kabsch, 1976) or SSM (Krissinel & Henrick, 2004). We analysed protein interfaces with PISA (Krissinel & Henrick, 2007). We used COILS for coiled-coil predictions (Lupas *et al*, 1991). For identification of homologous amino acid sequences, we used PsiBlast (Altschul *et al*, 1997) and the non-redundant National Center for Biotechnology Information protein database. We generated multiple amino acid sequence alignments (Fig EV5; Appendix Figs S3F and S5C and D) with TCoffee (Notredame *et al*, 2000) and formatted them with Esprit (Gouet *et al*, 1999) with the Blosum60 substitution matrix (Henikoff & Henikoff, 1992). For sequence alignments of yeast sequences of Ame1 or Okp1, and animal sequences of CENP-U or CENP-Q that we show in Fig EV7 and Appendix Fig S10, we used TCoffee with the PSI-Coffee option (Kemena & Notredame, 2009).

## Genetic modification, culturing and growth assays of *S. cerevisiae*

Most of our *S. cerevisiae* clones are derivatives of S288C type, unless specified otherwise in Table EV6. General *S. cerevisiae* genetic manipulation methods and media recipes for culturing and growth assays on solid medium were similar as described (Amberg *et al*, 2005). For our *Okp1_fl* native locus integration construct, we amplified, from *S. cerevisiae* genomic DNA, in one PCR a fragment of 250 bp of the 5′ untranslated region (UTR) immediately upstream of the coding region of *Okp1* and the coding region for Okp1; in a separate PCR, 250 bp 3′ UTR immediately downstream of *Okp1*; and from a pU6 plasmid a coding region for a 6×flag epitope. We assembled by PCR, from these fragments, a single DNA fragment, which has an NheI restriction enzyme site between the 3′ UTR and 5′ UTR sequences, that we ligated into plasmid pRS305 (Sikorski & Hieter, 1989), with isothermal assembly (Gibson *et al*, 2009), to generate plasmid *Okp1_fl-6×flag*-pRS305. We generated our *Okp1* integration constructs that lack coding regions for specific Okp1 segments by QuikChange-based site-directed mutageneses of *Okp1_fl-6×flag*-pRS305 with *PfuTurbo* DNA Polymerase AD or *PfuUltra* II Fusion Hot-Start DNA Polymerase (Agilent Technologies). For genomic integration in *S. cerevisiae*, we digested plasmids with NheI (New England BioLabs), transformed into diploid *S. cerevisiae* and selected single colonies that grew on solid agar with minimal synthetic complete medium without leucine. We sporulated clones in liquid sporulation medium and dissected tetrads on solid agar with yeast extract–peptone–dextrose (YPD) medium, with a dissection microscope (MSM, Singer Instruments). We constructed most of our clones of *S. cerevisiae* with specific genes removed or with genes (at native genomic locus) encoding C-terminal fluorescent protein fusion proteins by PCR-based methods and homologous recombination, as described (Longtine *et al*, 1998). For protein fusions with C-terminal GFP or C-terminal myc epitopes, we used plasmids pFA6a–GFP(S65T) kanMX6 or pFA6a–13×myc kanMX6 (Longtine *et al*, 1998), respectively, as templates for PCR products. For a complete *Ctf19* gene removal or removal of 5′ 954 base pairs of *Ctf19*, we used plasmid pRS303 (Sikorski & Hieter, 1989) as PCR template. Some of our clones, such as *Okp1_fl cnn1Δ* or *Okp1_cmΔ*

*cnn1Δ*, we generated by mating, sporulation in liquid medium and tetrad dissection. For a list of *S. cerevisiae* clones, see Table EV6. Clones are available upon request.

For analyses of spore viability of tetrad spores from *ctf19_1–954Δ/ctf19_1–954Δ*, *ctf19Δ/ctf19Δ*, *Okp1_fl/Okp1_fl* or *Okp1_cmΔ/Okp1_cmΔ*, *Okp1_cmΔnnΔ/Okp1_cmΔnnΔ* (Fig 8D; Appendix Fig S9G), we mated our respective isolated haploid clones, selected diploid clones on solid agar with minimal synthetic complete medium without adenine and without lysine, sporulated a selection of multiple single colonies in liquid medium and dissected tetrads from the sporulation culture. We calculated spore viability (Fig 8D) as percentage of the number of viable spores relative to the total number of dissected spores.

For vegetative growth assays on solid agar, we serially diluted our mitotically cycling *S. cerevisiae* clone suspensions to an absorbance at 600 nm ($A_{600}$) of 0.4, in 96-well plates (Nunc; Thermo Fisher Scientific). From this suspension, we sequentially fourfold diluted suspensions with minimal synthetic complete medium row-wise (Fig 8A and C; Appendix Fig S9A, B and E). We spotted *S. cerevisiae* suspensions with a sterilized 48-head metal pinner (V & P Scientific, Inc.). We prepared our solid YPD agar that contained benomyl with a final concentration of 20 μg/ml benomyl (Sigma-Aldrich). We did our growth experiments with *S. cerevisiae* clones that encode Mad1 fused to an auxin-inducible degron (Appendix Fig S9E) on solid yeast-extract peptone (YEP) agar with 2% (w/v) raffinose and 2% (w/v) galactose, and 1 mM of the synthetic auxin analogue 1-napthylic acetic acid (NAA; Carl Roth), similarly as described (Nishimura *et al*, 2009). To prepare *S. cerevisiae* extracts for our Western blot that we show an image in Appendix Fig S9F, we grew cultures in YEP with 2% (w/v) raffinose and 2% (w/v) galactose to an $A_{600}$ of 0.3. After the addition of NAA to a concentration of 1 mM, we collected cells from cultures after specific time points (Appendix Fig S9F) by centrifugation.

## Co-immunoprecipitation assays from *S. cerevisiae* extracts and Western blotting

For Western blots that we show images in Fig 7B and Appendix Fig S8B and C, we grew 50 ml cultures of *S. cerevisiae* clones at 30°C, to an $A_{600}$ of 0.7–2.3. For our co-immunoprecipitation experiment that we show Western blot images in Appendix Fig S8E and F, after growing our *S. cerevisiae* clone cultures to an $A_{600}$ of 0.4 at 30°C, we added nocodazole to a concentration of 15 μg/ml and continued to grow cultures for 2 h 15 min. After collecting *S. cerevisiae* cells by centrifugation, and storage at −80°C, we resuspended cells in 700 μl of a buffer with 25 mM HEPES pH 8.0, 150 mM NaCl, 5% (v/v) glycerol, 2 mM EDTA, 0.5 mM TCEP with protease inhibitors (Protease inhibitor cocktail set IV; Calbiochem) and phosphatase inhibitors. We lysed cells in a Minibeadbeater (BioSpec Products) at 4°C. We prepared cleared lysates by centrifugation in an Ultracentrifuge (Optima Max-XP, Beckmann Coulter), at 4°C. We coated protein G-coupled magnetic beads (Dynabeads, Thermo Fisher Scientific) with monoclonal M2 anti-flag antibodies (F1804, Sigma-Aldrich). We cross-linked antibodies to protein G with ~20 mM dimethyl pimelimidate in Na-borate pH 9.0. We blocked Dynabeads with bovine serum albumin (BSA). We incubated *S. cerevisiae* extracts with BSA-blocked Dynabead slurry, for ~16–18 h at 4°C. After washing Dynabeads in tubes on a magnet, we denatured

samples with SDS sample buffer at 95°C. We transferred proteins from SDS–PAGE gel onto a nitrocellulose membrane. We probed one membrane with monoclonal anti-flag M2 horseradish peroxidase (HRP)-coupled antibodies (from mouse; cat. no: A8592, Sigma-Aldrich). We probed a separate membrane with monoclonal 9e10 anti-myc antibodies (from mouse; Covance) and subsequently with HRP-coupled polyclonal anti-mouse antibodies (from goat; Jackson ImmunoResearch laboratories). We added ECL Western blotting detection reagent (GE Healthcare Life Sciences) as HRP substrate, and recorded chemiluminescence on high-performance chemiluminescence film (Amersham Hyperfilm ECL) or on a charge-coupled device of an Amersham Imager 600 (GE Healthcare Life Sciences). For our Western blot that we show an image of in Appendix Fig S9F, our protocol was similar (see our Appendix Supplementary Materials and Methods for details).

### Live cell fluorescence microscopy and image analysis

For our imaging, we grew *S. cerevisiae* clone cultures at 25°C or 30°C for ~16–18 h in YPD. For our clones with GFP fusion proteins, with this type of culture, we inoculated liquid minimal synthetic complete medium without tryptophan to an $A_{600}$ of ~0.4. In this medium, we grew mitotically cycling cells asynchronously for ~4–5 h at 30°C. We prepared cultures of our cells with Nuf2-mCherry or Bub1-3×GFP analogously in a similar medium with extra adenine (final concentration: 0.21% (w/v)). We imaged cells with *Okp1_fl* and Nkp1-GFP or Nkp2-GFP, or with *Okp1_nnΔ* and Nkp1-GFP or Nkp2-GFP (see Fig 5C), which we had immobilized on coverslips in glass bottom culture dishes (No. 0 coverglass, 0.085–0.13 mm; MatTek corporation) with concanavalin A, by confocal microscopy on an inverted Nikon Ti-E microscope equipped with an Andor AOTF laser combiner and a Yokogawa CSU-X1 spinning disc unit. We recorded fluorescence signals with an EMCCD (Andor Technology), and controlled acquisition with the Andor IQ3 software (Andor Technology). All our other living *S. cerevisiae* cells (see Fig 7C and D; Appendix Figs S8D and S9C and D) we imaged, after immobilization as described above, with a DeltaVision (Applied Precision, GE Healthcare Life Sciences) wide-field, inverted microscope with a Xenon Lamp and a Coolsnap HQ charge-coupled device (Photometrics). We controlled image acquisition and deconvoluted images with softWoRx software (DeltaVision Applied Precision). We analysed and processed our images with Fiji (Schindelin *et al*, 2012). Our live cell microscopy images are available upon request.

### Data accessibility for structural data

Our X-ray diffraction data are available from the Structural Biology Data Grid (https://data.sbgrid.org). Coordinates and structure factors for our crystal structure are available from the Protein Data Bank, with PDB accession code 5MU3.

**Expanded View** for this article is available online.

### Acknowledgements

We thank Stephen C. Harrison (Harvard Medical School) for support and funding for the early stage of our study; we are indebted to David S. King (Howard Hughes Medical Institute, University of California, Berkeley) for his mass spectrometry analyses; we thank Sheena D'Arcy (University of Texas, Dallas) and Ben E. Black (University of Pennsylvania) for the Excel macro for visualization of hydrogen-deuterium-exchange data; the NE-CAT beamline staff for assistance with X-ray diffraction data collection at APS (Argonne National Laboratory); Yoana N. Dimitrova, Stephen M. Hinshaw and Roberto Valverde (Harvard Medical School) for COMA-MIND purification, for *in vitro* translation constructs and purified Chl4-Iml3, and for help with sedimentation-equilibrium ultracentrifugation, respectively; SBGrid (Harvard Medical School) for computational support; Marcelo J. Berardi (formerly at Harvard Medical School) for help with electron microscopy and dynamic light scattering measurements; Pawel Pasierbek and Tobias Müller (Biooptics, Research Institute of Molecular Pathology (IMP), Vienna) for help with fluorescence microscopy; Arsen Petrovic (Max Planck Institute for Molecular Physiology, Dortmund) for analyses of sedimentation-equilibrium ultracentrifugation data; Pavel V. Afonine (Lawrence Berkeley National Laboratory) for advice on phenix.refine; Peggy Stolt-Bergner and Arthur Sedivy (Vienna BioCenter Core Facilities) for help with microscale thermophoresis experiments; Kim T. Simons (Emporia State University) for *S. cerevisiae* COMA construct; Alexander Schleiffer (IMP, Vienna) for help with amino acid sequence alignments; Gabriele Litos and Gülsah Pekgöz Altunkaya (IMP, Vienna) and Francesca Malvezzi (Astra Zeneca, Cambridge, UK) for help and discussions; Aleksandra Krolik and Shereen Kadir for help with graphical illustrations. F.S. received an APART fellowship (11428; Austrian Academy of Sciences); M.M.R. a Foundation for Polish Science PhD Grant; C.V.R. is supported by the Wellcome Trust [WT008150, WT099141]; M.D. by MAESTRO (2014/14/A/NZ1/0030) and Harmonia 5 grants (2013/10/M/NZ2/00298) from the Polish National Science Center; S.W. was funded by the Austrian Science Fund (FWF) and is funded by the German Research Foundation (DFG Grant WE 2886/2-1). Boehringer Ingelheim GmbH funds the IMP. For our research, we used resources of the Advanced Photon Source, a U.S. Department of Energy (DOE) Office of Science User Facility operated for the DOE Office of Science by Argonne National Laboratory under Contract No. DE-AC02-06CH11357.

### Author contributions

FS directed study, conceived and designed experiments, designed and cloned constructs, carried out protein purification, biochemical experiments, light scattering measurements, electron microscopy, sedimentation-equilibrium analytical ultracentrifugation measurements, yeast genetic modifications, co-immunoprecipitation experiments, live cell microscopy, protein crystallization, X-ray diffraction data collection, and crystal structure determination and analysed resultant data; MMR designed and carried out hydrogen-deuterium-exchange mass spectrometry experiments and analysed resultant data; YG designed and carried out nanoflow electrospray mass spectrometry experiments and analysed resultant data; FS, MMR and YG interpreted mass spectra; SW conceived and advised on yeast genetic experiments and carried out co-immunoprecipitation experiments and size-exclusion chromatography experiments for revised manuscript; FS wrote manuscript, with comments from MMR, YG, CVR, MD and SW. FS revised the manuscript. All authors approved the manuscript.

### Conflict of interest

The authors declare that they have no conflict of interest.

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
