## [Review Process File · The EMBO Journal]

Manuscript EMBO-2017-96636

Molecular basis for inner kinetochore configuration through RWD-domain interactions

Florian Schmitzberger, Magdalena M. Richter, Yuliya Gordiyenko, Carol V. Robinson, Michał Dadlez, and Stefan Westermann

Corresponding author: Florian Schmitzberger, University of Duisburg-Essen

Review timeline:

Submission date:	31 January 2017
Editorial Decision:	06 March 2017
Revision received:	31 July 2017
Accepted:	08 September 2017

Editor: Hartmut Vodermaier

Transaction Report:

1st Editorial Decision

06 March 2017

Thank you for submitting your manuscript on COMA complex architecture in the budding yeast outer kinetochores for our consideration. We have now received the comments of three expert referees, copied below for your information. As you will see, the referees appreciate the technical quality of your analyses and also agree on the importance of your results for the field, even if they are a bit divided regarding their views on the wider significance of the functional conclusions. Given these assessments, we shall be happy to consider a revised manuscript further for publication in The EMBO Journal, pending satisfactory addressing of the various experimental concerns raised by all three referees. Moreover, it will also be important to reorganize the manuscript and address the major presentational criticisms, keeping in mind the wider readership of our broad general journal.

Given that it is our policy to allow only a single round of major revision, I should stress that it will be important to carefully answer to all points raised at this stage. We generally allow three months as standard revision time, and it is our policy that competing studies published during this period will have no negative impact on our final assessment of your revised study; nevertheless I would appreciate if you contacted us as soon as possible upon publication of any related work, to discuss how to proceed.

Thank you again for the opportunity to consider this work for The EMBO Journal, and please do not hesitate to contact me should you have any comments or questions regarding the referee reports or this decision. I look forward to your revision.

REFEREE REPORTS

Referee #1:

The manuscript from Schmitzberger et al. describes the molecular interactions and organization of the budding yeast COMA complex. A structure of the Mcm21-Ctf19 heterodimer, previously determined by the same author, is shown bound to a section of Okp1. Using hydrogen-deuterium exchange, mass spectrometry and in vitro reconstitution the network of interactions within and around the COMA complex are defined. This has remained a rather ill-defined component of the kinetochore, and this work complements recent advances in our understanding of the so-called CCAN and KMN networks. The study is comprehensive and the mass-spectrometry and structural work look solid. I do have some concerns about the reconstitution experiments which the authors should address. Overall, think the work will be of considerable interest to the kinetochore community, but may be rather too specialized for the wider EMBO J. readership, especially as the functional insights are limited.

Major Points:

1. The COMA-Nkp1/Nkp2 interaction studies do not seem to be consistent. From the hydrogen-deuterium exchange, SEC experiments and reconstitution presented in figure 3, the authors conclude that the C-termini of Okp1 and Ame1 are required to bind Nkp1/Nkp2. However, in the limited proteolysis experiment shown in figure S2H, the peak elution contains just Mcm21 and Ctf19 with Nkp1 and Nkp1 still bound in stoichiometric quantities. Where are the Okp1 and Ame1 fragments? If the proteins have been fully proteolysed, it is hard to see how this can be reconciled with Nkp1/Nkp2 binding to their C-termini. If it is possible that a short fragment of Okp1 and Ame1 are protected and remain in the elution peak to link Ctf19/Mcm21 to Nkp1/2 this needs to be identified. A useful control experiment would also be mixing the Ctf19/Mcm21 heterodimer with Nkp1/Nkp2 and checking for interaction by SEC.
2. The Ame1/Okp1 reconstitution experiments presented in figure 3C are not convincing. In several cases (e.g. lanes 1,3,4) the binding appears sub-stoichiometric, as judged by band intensity, making it difficult to draw robust conclusion about interacting sections. Also, what are the unlabeled bands between lanes 5 and 6?
3. The section on the live cell imaging of the Okp1_{cm} truncation is extremely confusing (page 15). The authors state they are combining the Okp1 mutant with mutants that lack non-essential CTF19 subunits, then talk about a Okp1_{cm}Δ cnn1Δ double mutant. Firstly, what does CTF19 (uppercase) refer to - the Ctf19 protein, the Ctf19/Mcm21 heterodimer, or the COMA complex? Secondly, as far as I am aware, Cnn1 is not a CTF19 subunit. It is not clear to me that there should be any dependence on Cnn1 for CTF19 (whatever this means) localization. Also, when the authors talk about "absence of multiple CTF19 subunits" - what are they referring to? This section needs to be substantially re-written, and a clear explanation for the effect of the double mutant to be provided.

Minor Points:

1. The presentation of the study could be improved. The main figures (particularly 1-3) display large amounts of data, much of it derived from mass spectrometry or size-exclusion runs but the key point of the figure is often hard to discern. The writing would also benefit from being more concise in places. I think paying some attention to both these aspects (e.g. by moving raw MS spectra to supplementary materials and simplifying main figures) would make the manuscript considerably more accessible to both specialist and non-specialist readers.
2. Introduction - page four. The authors claim to have determined the Ctf19-Mcm21-Okp1 crystal structure. The structure presented includes 24 residues from Okp1. This statement should be revised.
3. Pages 5-6. The observation that Nkp1 & Nkp2 dimerize has been previously reported (Brooks et al., Structure 2010). A genetic interaction and between Nkp2 and COMA, and co-localization dependence has been described (Tirupataiah et al., Mol. Biol. Reports, 2014).
4. Page 6 - To those unacquainted with biophysics, it is not clear why a smaller r.m.s. radius than hydrodynamic radius implies flexibility. Neither of these necessarily relate to measurements made on EM micrographs either. The section should be made clearer.

5. Page 6 - "Okp1 N-terminal and C terminal regions (residues 1-164 and 330-383) are contiguously lacking a stable hydrogen-bonding network." It is not clear what this statement means.

6. Figure 3A - what do the thicker lines on the linear protein schematics represent?

7. Page 11 - Phe329 of Okp1 is suggested as mediating a specific interaction with Ctf19. This residue is not indicated in figure 4D. It is also described as being conserved in budding yeasts, but in the alignments shown, it is only present in the *K. lactis* protein.

8. The authors need to show an unbiased Fo-Fc omit map at a stated contour level for the Okp1 peptide in their structure.

9. The authors state that the rapid evolution of centromere proteins is due to some type of genetic drive (page 16 and associated reference). At present, this is a speculative idea that explains some features of centromeres, but is unproven, and should not be presented as fact. This comment could be dropped or at least moderated.

10. The finding the COMA proteins contribute to checkpoint signaling in the absence of Aurora B activity has been previously described (Matson et al., *Genes & Dev.* 2012), and should be included in the discussion.

Referee #2:

The authors examine the structure of budding yeast COMA, a heterotetrameric inner kinetochore complex, and the interaction of COMA with associated proteins. COMA consists of two essential components (Okp1 and Ame1) that localize the complex to the kinetochore and are required for the establishment of the outer kinetochore. The two other components (Ctf19 and Mcm21) and COMA associated proteins (defined mostly by co-purification with COMA) are non-essential. They are known to increase chromosome segregation fidelity. Furthermore, corresponding deletion mutants show synthetic growth defects with spindle assembly checkpoint (SAC) mutants.

The authors apply state of the art methods to address how COMA subunits interact with each other and associated proteins. These include: The collision-induced destabilization of isolated complexes and subsequent identification of subcomplexes by mass spectrometry to gain insight which components interact directly. H/D exchange experiments that provide strong indications for protein interaction sites. In vitro reconstitution of complexes with deletion constructs to support the sites. X-ray analysis of a trimeric Okp1/Ctf19/Mcm21 minimal construct. Isothermal titration-calorimetry and microscale thermophoresis to determine the K_d and specificity of the observed Okp1/Ctf19/Mcm21 interaction. In summary, these approaches reveal Okp1 to function as a hub within the inner kinetochore. Okp1 strongly interacts with the Ctf19/Mcm21 heterodimer via a short peptide sequence that inserts between the (previously described) D-RWD domains of the Ctf19/Mcm21 heterodimer. Ame1 binds to an independent segment of Okp1 but does not directly interact with Ctf19/Mcm21. A third region located at the C-terminus of Okp1 binds a heterodimer of Nkp1/2 (two of the COMA associated proteins) that also interacts with a C-terminal region in Ame1. A central "core" region of Okp1 can be speculated to confer the interaction of Okp1 with Mif2. The authors support some of their conclusions by showing that the deletion of the Ctf19/Mcm21 binding domain in Okp1 results in the expected phenotypes in vivo (defect in kinetochore localization of Ctf19 but not Nkp1/Nkp2 and SAC requirement for efficient growth).

The technical quality of the experiments performed is high and the conclusions drawn are convincing for most parts. The work presented is a substantial advance in the understanding of the organization and possible dynamics within the inner kinetochore of budding yeast. Since work on the *S. cerevisiae* kinetochore has clearly contributed strongly to the current knowledge on kinetochores and chromosome segregation, I consider the presented findings important and interesting for the field (despite that it is currently unclear how much can be applied to higher Eukaryotes).

A few points should be addressed:

1) I find the result section hard to read for several reasons:

a) It is often organized according to the applied techniques and not according to the individual protein interactions examined. For instance, we learn in the second chapter that Nkp1-Nkp2 binds to Ame1-Okp1 and in the third chapter that deuterium exchange suggests that it binds to the C-termini of Okp1 and Ame1. Then the topic is left in the fourth chapter and in the fifth the interaction sites are confirmed by reconstitution experiments. Similar is true for the Ame1/Okp1 and Okp1/Ctf19/Mcm21 interaction.

b) The headlines of the chapters are often very general and lack focus, for instance "Definition of inner kinetochore assembly-requirements" or "Molecular basis for Okp1 interactions at the inner kinetochore"

c) There are innumerable links to the supplement. Some of the data in the supplement could be shifted to the main section. For instance, data presented in Fig. S2A-C that supports the Nkp1-Nkp2 interaction with Okp1-Ame1. To gain space I would condense Fig 5 and 6 that contain predominantly confirmatory data that have in principle been published elsewhere.

2) The conclusion drawn in chapter 5 of the result section that Nkp1 interacts with Okp1 and Nkp2 with Ame1 is not based on data shown in this manuscript. For Okp1-Nkp1 the authors refer to "data not shown" elsewhere in the manuscript and for Ame1-Nkp2 to Fig S2B. I was however unable to find evidence for the occurrence of a Ame1-Nkp2 subcomplex in this mass spectrum, although there is a complex labeled Ctf19-Mcm21-Okp1-HisNkp1-Nkp1 that probably should read-Nkp2. The postulated interactions thus are not supported by strong data and should be moved from the results to the discussion section.

3) The authors find that COMA associated proteins as Ctf3, Mcm22, Mcm16 still localize to kinetochores in vivo when the Ctf19/Mcm21 binding site in Okp1 is deleted (and Ctf19 and Mcm21 are thus absent from the kinetochore). They conclude from this that these proteins do not interact with Ctf19/Mcm21 but bind to Okp1/Ame1 via Mif2 (as shown in Fig. 5E). Given the discrepancy with former ChIP results in this respect, I would be careful making this conclusion. Ctf3 ect. could interact with Ctf19/Mcm21 but this interaction might not be absolutely required for kinetochore localization (i.e. they are localized also via other interactions). Since the interaction of COMA with Ctf3/Mcm16/Mcm22 was originally defined by affinity co-purification, I strongly suggest that the authors test their model by analyzing the co-purification (co-IP) of Ctf3 with Okp1/Ame1 in the presence and absence of Ctf19/Mcm21.

Referee #3:

The kinetochore is a large protein complex, which connects chromatin to microtubule for faithful chromosome segregation. The kinetochore is composed of various proteins, which are conserved between yeast and human. Although many studies revealed composition and organization of the kinetochore in various organisms, detail structural organization has not been fully understood yet. In this paper, authors want to clarify structural organization of yeast COMA complex, which contains Ame1, Ctf19, Mcm21, and Okp1. Their analyses suggest that Okp1 functions as a hub for the complex assembly: Okp1 multiple regions bind to various proteins distinctly, based on various structural and biochemical assays. They also found that the COMA complex associates with Nkp1-Nkp2 and Chl4-Ime1 complexes. Finally, they demonstrate that the Okp1 motif for binding to Ctf19-Mcm21 is required for kinetochore function in the absence of mitotic checkpoint.

This is a well-done study and is an important step to understand yeast kinetochore architecture. I would like to suggest several points to improve quality of this paper. However, I believe that authors could address these points smoothly. My specific points are followings.

1. Concerning terminology, authors are using "COMA" as a complex containing Ame1, Ctf19, Mcm21, and Okp1. Then, they are also using "CTF19" as a larger complex including COMA. For

general readers, it is hard to distinguish "Ctf19" and "CTF19". I suggest that they should use another word for "CTF19" like CCAN.

2. On page 5, authors state "we observed dimeric COMA subassemblies without Ctf19, Mcm21, or Ctf19-Mcm21 but none without Ame-Okp1 (Fig. S1C). We conclude that COMA dimerizes through Ame-Okp1." It is a bit unclear about this conclusion, because they found both Ame1-Okp1 and Ctf19-Mcm21 dimers. COMA may be dimerized through Ctf19-Mcm21?

3. On page 7, authors conclude that Okp1 segment3 and Ame1 segment1 bind Nkp1-Nkp2. Although I agree that several experiments suggest this, it may be better to show direct evidence for this conclusion using recombinant proteins.

4. For in vivo analysis, they just show phenotype for cells expressing Okp1 lacking segment1. It may be better show other mutants such Okp1 lacking segment 2. This should lead to Ame1 mislocalization. They can also test Okp1 mutant lacking segment 3.

5. In addition, it is a bit unclear how Chl4-Ime3 is associated with Ctf19-Mcm21. Do they think that Chl4-Ime3 directly binds to Ctf19-Mcm21? Please clarify this.

6. On page 15, authors state "On solid medium, ... non-essential CTF19 subunits." This should be stated on page 13. On page 13, although readers can understand protein localization, it is unclear about consequence for chromosome segregation in cells expressing Ctf19 cm.

1st Revision - authors' response

31 July 2017

We thank the three reviewers for their comments and helpful suggestions that aided us to improve our manuscript. We appreciated to see that all reviewers recognized the general relevance of our manuscript for the kinetochore field, and the depth of our study. We were pleased to read that our presented work is 'a substantial advance in the understanding of the organization and possible dynamics within the inner kinetochore' (Reviewer 2), that it is 'an important step to understand yeast kinetochore architecture' (Reviewer 3), and that our study is 'comprehensive and the mass spectrometry and structural work look solid' (Reviewer 1).

Through additional experiments, which substantiate and extend our previous conclusions, we have addressed suggestions or concerns from the reviewers. Our revised manuscript version includes the following new experimental results.

1. We generated several Okp1-mutant alleles (of *S. cerevisiae*) that each lack one (or two) of our identified structured Okp1 segments, and we characterized their phenotypes. We found that cells without Okp1 segment 2 or Okp1 core are not viable, consistent with our suggestion that these segments make protein-protein interactions with essential kinetochore subunits—Ame1 or Mif2, respectively. Mutants that either lack the Nkp1-Nkp2 binding motif, or both the Nkp1-Nkp1 binding motif and the Ctf19-Mcm21 binding motif are viable (again as we expected). We demonstrate with fluorescence microscopy, that in living cells, absence of Okp1 segment 3 abrogates kinetochore localization of Nkp1-Nkp2, consistent with our biochemical data.
2. We have established, by size-exclusion chromatography with recombinant samples, that Ame1-Okp1—on its own—suffices to bind Nkp1-Nkp2; while Ctf19-Mcm21 does not bind Nkp1-Nkp2.
3. In our co-immunoprecipitation experiments with extracts from cells with Mcm16 (a subunit of the CTF3 complex) and either Okp1 or Okp1_{cmΔ}, we found that Mcm16 does not co-immunoprecipitate with Okp1_{cmΔ} (while Mcm16 does localize to kinetochores in living cells with Okp1_{cmΔ}). Our new experiments establish differential association dependencies (on the Ctf19-Mcm21 binding motif) between COMA and the CTF3-complex subunits in yeast extracts *versus* living yeast cells.
4. We report sedimentation-equilibrium ultracentrifugation data, to confirm that Ctf19-Mcm21—on its own—is monomeric in solution, and that COMA at low micromolar concentrations, is in a monomer-dimer equilibrium in solution—in agreement with our data that we reported in our initially submitted manuscript.

In response to the criticism concerning the general organization of our manuscript and accessibility to a wider audience, we have re-structured and re-written several of our text sections. We simplified several of our multi-panel figures, by splitting previously condensed panels up into different figures, and introduced additional figures. We also added annotations in our figures. As suggested by the reviewers, we have improved the order of presentation of our data. We now, for instance, introduce early on in our manuscript the growth phenotype of our different *S. cerevisiae* mutants that lack specific Okp1 segments. In the following, we detail our responses to the specific points raised by the reviewers.

Responses to reviewer #1

Major Points:

1. The COMA-Nkp1/Nkp2 interaction studies do not seem to be consistent. From the hydrogen-deuterium exchange, SEC experiments and reconstitution presented in figure 3, the authors conclude that the C-termini of Okp1 and Ame1 are required to bind Nkp1/Nkp2. However, in the limited proteolysis experiment shown in figure S2H, the peak elution contains just Mcm21 and Ctf19 with Nkp1 and Nkp1 still bound in stoichiometric quantities. Where are the Okp1 and Ame1 fragments? If the proteins have been fully proteolysed, it is hard to see how this can be reconciled with Nkp1/Nkp2 binding to their C-termini. If it is possible that a short fragment of Okp1 and Ame1 are protected and remain in the elution peak to link Ctf19/Mcm21 to Nkp1/2 this needs to be identified. A useful control experiment would also be mixing the Ctf19/Mcm21 heterodimer with Nkp1/Nkp2 and checking for interaction by SEC.

We thank the reviewer for raising this important issue and for looking at our data in detail. We have included a more representative SDS-PAGE gel image in Fig EV4 (our initial gel image was indeed misleading), which shows that proteolytic fragments of Okp1 are co-eluting with Ame1 fragments, Ctf19-Mcm21, and Nkp1-Nkp2. We have also indicated in our new figure the SEC fractions that we pooled and used for high-resolution mass spectrometry analysis. We have included a table with the Okp1 (and other protein) fragments that we identified in our mass-spectrometry analysis and the molecular weights of these fragments. We previously presented these fragment data in our supplementary tables 1 and 2, where they were more difficult to associate with the SDS-PAGE image. Our mass spectrometry data show clearly that Okp1 fragments in the main SEC elution peak contain all three binding sites—those for Ame1, Ctf19-Mcm21, and Nkp1-Nkp2. Such Okp1 fragments are with residues 111–383 or 117–383. We hope that by showing this information more prominently in an expanded view figure, it will be more accessible.

We had already in Supplementary Fig 2D of our initially submitted manuscript version presented a SEC experiment of Ctf19-Mcm21 and Nkp1-Nkp2. These data are now shown (more prominently) in Fig EV3A. Ctf19-Mcm21 and Nkp1-Nkp2 do not co-elute—consistent with the absence of interactions between Ctf19-Mcm21 and Nkp1-Nkp2, as suggested by our mass-spectrometry data. We now have also included SEC data that show that recombinant samples of Ame1-Okp1 and Nkp1-Nkp2 co-elute (Fig EV3B), again in agreement with our mass spectrometry data and presented binding data.

2. The Ame1/Okp1 reconstitution experiments presented in figure 3C are not convincing. In several cases (e.g. lanes 1,3,4) the binding appears sub-stoichiometric, as judged by band intensity, making it difficult to draw robust conclusion about interacting sections. Also, what are the unlabeled bands between lanes 5 and 6?

It is rather common to find the His-tagged component, in our case Okp1 variants, in excess to the co-purifying complex component/s, in our case Ame1, on SDS-PAGE gels from Ni²⁺ affinity-purifications (of bacterially produced protein complexes). This is consistently the case for all our affinity purifications with His-tagged Okp1 variants that we show in Fig 4C. Likewise, histidine-tagged Mcm21 (His-Mcm21) is present in excess of the other COMA components in our COMA affinity purifications that we show a gel of in Fig 4B. We have purified most of our Ame1-Okp1 variants also by ion-exchange and size-exclusion chromatography—subsequent to affinity chromatography. To clarify the stoichiometric presence of Ame1 and Okp1 in our purifications, we have included in Appendix Fig S4, images of SEC graphs and SDS-PAGE gel-images from SEC

fractions, from representative purifications of two of our Ame1-Okp1 variants. These SDS-PAGE gel-images show stoichiometric presence of Ame1 and Okp1 co-eluting from SEC. We have also added labels to Fig 4C that indicate that Okp1 variants are His-tagged.

The unlabeled lanes can be ignored. We have clarified this in the figure legend. We chose not to splice these lanes, but show the whole SDS-PAGE instead, because previous reviewers advised against doing so.

3. The section on the live cell imaging of the Okp1_cm truncation is extremely confusing (page 15). The authors state they are combining the Okp1 mutant with mutants that lack non-essential CTF19 subunits, then talk about a Okp1_cmΔ cnn1Δ double mutant. Firstly, what does CTF19 (uppercase) refer to - the Ctf19 protein, the Ctf19/Mcm21 heterodimer, or the COMA complex? Secondly, as far as I am aware, Cnn1 is not a CTF19 subunit. It is not clear to me that there should be any dependence on Cnn1 for CTF19 (whatever this means) localization. Also, when the authors talk about "absence of multiple CTF19 subunits" - what are they referring to? This section needs to be substantially re-written, and a clear explanation for the effect of the double mutant to be provided.

We have re-phrased and clarified this section. To avoid confusion, we now refer to CCAN instead of CTF19 throughout our manuscript. With our previous statement 'absence of multiple CTF19 subunits', which we have now re-phrased, we meant the absence of non-essential CCAN subunits in addition to Ctf19-Mcm21 and Chl4-Iml3, which are delocalized in the absence of the Ctf19-Mcm21 binding motif (in Okp1_cmΔ). Testing the absence of subunits from multiple 'branches' of CCAN subunits seemed a plausible experiment for us to do. We cannot, at present, offer an unambiguous explanation for the enhanced growth defect of our Cnn1 Okp1_cmΔ double mutant, compared with the individual mutants. The specific functional relevance of Cnn1 is unclear (see Malvezzi et al., EMBO J, 2013). It appears redundant. But frequently, a Cnn1 deletion exhibits phenotypes in combination with other gene deletions.

Minor Points:

1. The presentation of the study could be improved. The main figures (particularly 1-3) display large amounts of data, much of it derived from mass spectrometry or size-exclusion runs but the key point of the figure is often hard to discern. The writing would also benefit from being more concise in places. I think paying some attention to both these aspects (e.g. by moving raw MS spectra to supplementary materials and simplifying main figures) would make the manuscript considerably more accessible to both specialist and non-specialist readers.

We have re-organized several of our figures, and split data up into different (new) figures, to simplify the display of information. We have re-organized and shortened the text of our first two paragraphs, and clarified sections. We have left some of the annotated mass spectra in our main figures, because we feel they are important to follow the main manuscript, and because reviewer #2 asked to see some of the mass-spectrometry data previously in our supplementary figures in the main figures instead.

2. Introduction - page four. The authors claim to have determined the Ctf19-Mcm21-Okp1 crystal structure. The structure presented includes 24 residues from Okp1. This statement should be revised.

We changed it to 'crystal structure of Ctf19-Mcm21 with interacting Okp1 segment'.

3. Pages 5-6. The observation that Nkp1 & Nkp2 dimerize has been previously reported (Brooks et al., Structure 2010). A genetic interaction and between Nkp2 and COMA, and co-localization dependence has been described (Tirupataiah et al., Mol. Biol. Reports, 2014).

We included the reference to Brooks et al. 2010. Brooks et al report what we understand is an indication for an interaction between Nkp1 and Nkp2, based on visual inspection of SDS-PAGE gel-images of affinity-purification fractions from yeast extracts. We could, however, not convince ourselves of these data, because the link to the images in the supplementary section of the article does not seem to be functional anymore. We chose not to cite Tirupataiah et al. 2014, because we found it difficult to evaluate the quality of the reported data in that manuscript. This manuscript reports a kinetochore localization dependence of Nkp2 on Ctf19 and Mcm21 (based on

immunofluorescence of (dead) cells). We found this result to be inconsistent both with our in vivo imaging data and our biochemical data. We show with our new data (from live cell microscopy) that kinetochore localization of Nkp1-Nkp2 depends on Okp1 segment 3, consistent with our biochemical data.

4. Page 6 - To those unacquainted with biophysics, it is not clear why a smaller r.m.s. radius than hydrodynamic radius implies flexibility. Neither of these necessarily relate to measurements made on EM micrographs either. The section should be made clearer.

We rephrased this part, and removed the reference to the r.m.s.d. radius.

5. Page 6 - "Okp1 N-terminal and C terminal regions (residues 1-164 and 330-383) are contiguously lacking a stable hydrogen-bonding network." It is not clear what this statement means.

We corrected this sentence, by removing the term 'contiguous', which was misleading. Deuterium-exchange + mass spectrometry measures the degree of hydrogen bonding. Fast deuterium-exchange implies absence of stable hydrogen bonding network, and a disordered or flexible state.

6. Figure 3A - what do the thicker lines on the linear protein schematics represent?

The thick(er) lines represent the structured segments that we identified with our deuterium-exchange experiment that we show in Fig 3A,B. We added an explanation for the annotation to our figure legend.

7. Page 11 - Phe329 of Okp1 is suggested as mediating a specific interaction with Ctf19. This residue is not indicated in figure 4D. It is also described as being conserved in budding yeasts, but in the alignments shown, it is only present in the *K. lactis* protein.

We corrected our writing. We added a label for Phe329 (Fig 6D). A bulky hydrophobic residue, phenylalanine or leucine, is conserved. Depending on the used (amino-acid) substitution matrix, this is a conserved change.

8. The authors need to show an unbiased Fo-Fc omit map at a stated contour level for the Okp1 peptide in their structure.

We included an image of an mFo-DFc electron-density map (at 2.7 σ)—unbiased for the co-crystallized Okp1 fragment—in Appendix Fig S7C.

9. The authors state that the rapid evolution of centromere proteins is due to some type of genetic drive (page 16 and associated reference). At present, this is a speculative idea that explains some features of centromeres, but is unproven, and should not be presented as fact. This comment could be dropped or at least moderated.

We moderated the statement, as suggested; and we added the original reference to 'centromeric drive' from Henikoff 2001. At least in animals, centromeric drive is a plausible concept explaining the rapid evolution of centromere-associated proteins.

10. The finding the COMA proteins contribute to checkpoint signaling in the absence of Aurora B activity has been previously described (Matson et al., Genes & Dev. 2012), and should be included in the discussion.

We included this reference in our discussion section.

Responses to reviewer #2

A few points should be addressed:

1) I find the result section hard to read for several reasons:

a) It is often organized according to the applied techniques and not according to the individual protein interactions examined. For instance, we learn in the second chapter that Nkp1-Nkp2 binds to Ame1-Okp1 and in the third chapter that deuterium exchange suggests that it binds to the C-termini of Okp1 and Ame1. Then the topic is left in the fourth chapter and in the fifth the interaction sites are confirmed by reconstitution experiments. Similar is true for the Ame1/Okp1 and Okp1/Ctf19/Mcm21 interaction.

*We thank the reviewer for these suggestions. We restructured several sections. Our revised manuscript version now starts with a description of the identification of the COMA-Nkp1-Nkp2 assembly, its biophysical characterization, followed by the definition of Okp1 binding sites for Nkp1-Nkp2 and for Ctf19-Mcm21; followed by a description of our crystal structure. In our revised Fig 5, we have combined biochemical experiments and live cell fluorescence-imaging to characterize the Nkp1-Nkp2 binding site in Okp1. We introduce the phenotype of our *S. cerevisiae* variants, which lack specific structured segments of Okp1, on page 7, and thus early in our manuscript. We hope that the logic of our manuscript is now easier to follow, including for non-specialized reader. We did, however, not find a way to introduce our Nkp1-Nkp1 in vitro binding experiments earlier, because those descriptions depend on knowledge of the description of our recombinant COMA variants.*

b) The headlines of the chapters are often very general and lack focus, for instance "Definition of inner kinetochore assembly-requirements" or "Molecular basis for Okp1 interactions at the inner kinetochore"

We made our section headers more COMA-Nkp1-Nkp2 specific. For example, we now begin with 'Identification of a COMA-Nkp1-Nkp2 assembly and its molecular composition'; and we have changed the mentioned headers to 'Definition of COMA assembly requirements', and 'Molecular basis for Okp1 interaction in COMA'.

c) There are innumerable links to the supplement. Some of the data in the supplement could be shifted to the main section. For instance, data presented in Fig. S2A-C that supports the Nkp1-Nkp2 interaction with Okp1-Ame1. To gain space I would condense Fig 5 and 6 that contain predominantly confirmatory data that have in principle been published elsewhere.

We re-organized our figures and the general figure presentation by separating some supplementary figures in enhanced view figures or appendix supplementary figures. The former are more relevant to follow the main manuscript figures and text, the latter are for a more specialized audience. We hope that, in doing so, we have reduced the complexity of our presented data. We moved all of our deuterium-exchange data time courses to the Appendix supplementary figures. As suggested, we have also moved some of our mass spectra of COMA-Nkp1-Nkp2 from our previous supplementary figure to our main figure 1. We re-organized Fig 1 and Fig 2. We shifted data to a figure—Fig 3.

2) The conclusion drawn in chapter 5 of the result section that Nkp1 interacts with Okp1 and Nkp2 with Ame1 is not based on data shown in this manuscript. For Okp1-Nkp1 the authors refer to "data not shown" elsewhere in the manuscript and for Ame1-Nkp2 to Fig S2B. I was however unable to find evidence for the occurrence of a Ame1-Nkp2 subcomplex in this mass spectrum, although there is a complex labeled Ctf19-Mcm21-Okp1-HisNkp1-Nkp1 that probably should read ...-Nkp2. The postulated interactions thus are not supported by strong data and should be moved from the results to the discussion section.

We have clarified this part, and moved the speculative parts to our discussion. Our description of a mass for Ctf19-Mcm21-Okp1-HisNkp1-Nkp1 is, however, correct (our sample contained a mixture of His-Nkp1 and Nkp1).

3) The authors find that COMA associated proteins as Ctf3, Mcm22, Mcm16 still localize to kinetochores in vivo when the Ctf19/Mcm21 binding site in Okp1 is deleted (and Ctf19 and Mcm21 are thus absent from the kinetochore). They conclude from this that these proteins do not interact with Ctf19/Mcm21 but bind to Okp1/Ame1 via Mif2 (as shown in Fig. 5E). Given the discrepancy with former ChIP results in this respect, I would be careful making this conclusion. Ctf3 ect. could interact with Ctf19/Mcm21 but this interaction might not be absolutely required for kinetochore localization (i.e. they are localized also via other interactions). Since the interaction of COMA with

Ctf3/Mcm16/Mcm22 was originally defined by affinity co-purification, I strongly suggest that the authors test their model by analyzing the co-purification (co-IP) of Ctf3 with Okp1/Ame1 in the presence and absence of Ctf19/Mcm21.

We thank the reviewer for this suggestion. We have, as suggested, done co-immunoprecipitation (Co-IP) experiments from S. cerevisiae extracts, with the CTF3 complex subunit Mcm16 and our (wildtype) Okp1 or Okp1_cmΔ variant. We show the data of our experiments in Appendix Fig S8E,F. From metaphase-arrested cells, a relatively low amount of Mcm16 co-immunoprecipitated with Okp1, but did not co-immunoprecipitate with Okp1_cmΔ. This result is consistent with related experiments that showed that Ctf3 does not co-immunoprecipitate with Ame1-Flag from Ctf19Δ cells (Pekgoz Altunkaya et al., 2016). We have tested for interaction with recombinant, bacterially produced samples of S. cerevisiae COMA and S. cerevisiae Ctf3-Mcm16-Mcm22 (CTF3 complex) with SEC. We could not detect an interaction between these assemblies (in the absence of other kinetochore proteins or DNA). Our observation reinforces the general notion that association dependencies determined with Co-IP experiments from soluble extracts can differ substantially from those determined with live cell imaging. We suggest that chromatin binding, presumably mediated by histone-fold domains of proteins in the Ctf3-Cnn1-Wip1 complex, contribute to localization of kinetochore subunits to centromeres, and thus CCAN assembly. Conditions for chromatin binding are probably often not maintained in soluble yeast-cells extracts, after their preparation from lysed cells. We have added clarifying sentences to our results section and discussion section.

Responses to reviewer #3

1. Concerning terminology, authors are using "COMA" as a complex containing Ame1, Ctf19, Mcm21, and Okp1. Then, they are also using "CTF19" as a larger complex including COMA. For general readers, it is hard to distinguish "Ctf19" and "CTF19". I suggest that they should use another word for "CTF19" like CCAN.

We corrected accordingly. We now explain in our introduction that we use the term CCAN for CTF19 throughout our manuscript.

2. On page 5, authors state "we observed dimeric COMA subassemblies without Ctf19, Mcm21, or Ctf19-Mcm21 but none without Ame-Okp1 (Fig. S1C). We conclude that COMA dimerizes through Ame-Okp1." It is a bit unclear about this conclusion, because they found both Ame1-Okp1 and Ctf19-Mcm21 dimers. COMA may be dimerized through Ctf19-Mcm21?

We have clarified this paragraph, and provide additional data in our revised manuscript version. We had previously reported that Ctf19-Mcm21, on its own, does not dimerize (see Schmitzberger&Harrison, EMBO Reports, 2012). Even at the high protein concentration in our crystals, we did not find indications for Ctf19-Mcm21 dimerization. We have in our revised manuscript version, included analytical sedimentation-equilibrium ultracentrifugation data that we measured of Ctf19-Mcm21 or COMA. We show these data in Appendix Fig S1C,D. Our analysis shows that Ctf19-Mcm21 up to concentrations of 16 μM is monomeric in solution. We are thus confident that Ctf19-Mcm21—on its own—does not dimerize, but that it is the Ame1-Okp1 moiety that accounts for COMA dimerization.

3. On page 7, authors conclude that Okp1 segment3 and Ame1 segameny1 bind Nkp1-Nkp2. Although I agree that several experiments suggest this, it may be better to show direct evidence for this conclusion using recombinant proteins.

We have, as suggested, done SEC experiments with recombinant samples; and we show, in Fig EV2B, co-elution of S. cerevisiae Ame1-Okp1 with S. cerevisiae Nkp1-Nkp2, as we expected. We could not produce substantial amounts of recombinant full-length K. lactis Ame1-Okp1 (as we previously stated in our supplementary information part). We have, during our manuscript revision, attempted to produce in bacteria a number of truncated versions of K. lactis Ame1-Okp1 (similar to the constructs that we show in Fig 4C) that either have intact C termini or lack one or both of the two C termini. Most of these constructs did not produce in substantial soluble amounts. We now show, however, by fluorescence microscopy with Nkp1-GFP or Nkp2-GFP that Okp1 segment 3 is required for Nkp1-Nkp2 kinetochore localization in living S. cerevisiae cells (Fig 5C).

4. For in vivo analysis, they just show phenotype for cells expressing Okp1 lacking segmanet1. It may be better show other mutants such Okp1 lacking segment 2. This should lead to Ame1 mislocalization. They can also test Okp1 mutant lacking segment 3.

We have prepared S. cerevisiae mutant clones that lack one (or two) of the structured Okp1 segments. We found that clones without Okp1 segment 2 (Ame1 binding site) or Okp1 core (possible Mif2 binding site) are not viable; as we expected for an interaction of these segments with the essential kinetochore proteins Ame1 or Mif2 (,which our biochemical data suggest). We show that a mutant (Okp1_{nnΔ}) that lacks segment 3 (the Nkp1-Nkp2 binding site) is viable; but that localization of Nkp1-GFP and Nkp2-GFP to kinetochores is abrogated in this mutant—consistent with our biochemical data. A mutant that lacks both the Nkp1-Nkp2 binding site and Ctf19-Mcm21 binding site (Okp1_{cmΔnnΔ}) is also viable—as we expected, because in this mutant again only non-essential CCAN subunits are absent from mitotic kinetochores. We also show that Okp1_{cmΔnnΔ} does not have a pronounced meiotic phenotype (Fig 8D).

5. In addition, it is a bit unclear how Chl4-Ime3 is associated with Ctf19-Mcm21. Do they think that Chl4-Ime3 directly binds to Ctf19-Mcm21? Please clarify this.

We thank the reviewer for pointing out to us that this was unclear. We clarified this point and added several sentences addressing it in our discussion section. Our binding experiments (Fig EV6) suggest that Chl4-Iml3 probably interacts with Ctf19-Mcm21 through Chl4. It is currently not clear to us, why we failed to detect an interaction on SEC with bacterially produced Chl4-Iml3 and Ctf19-Mcm21 samples. We speculate that a stable interaction may require binding contributions from other CCAN subunits, such as Mif2, and/or posttranslational modifications. An interaction of Chl4 may be with the parts N terminal of the Ctf19-Mcm21 D-RWD domains, which contain conserved residues and are disordered in our crystals (see Schmitzberger & Harrison, 2012). The D-RWD domains themselves have few conserved surface residues, outside of the Okp1 interaction site.

6. On page 15, authors state "On solid medium, ... non-essential CTF19 subunits." This should be sated on page 13. On page 13, although readers can understand protein localization, it is unclear about consequence for chromosome segregation in cells expressing Ctf19 cm.

We agree. In our revised manuscript version, we mention already on page 7, when introducing yeast mutants that lack specific Okp1 segments, that Okp1_{cmΔ} or Okp1_{nnΔ} mutant clones are viable (while mutants that lack Okp1 segment 2 or core are not viable); and thus that—by extension—the chromosome segregation rates of Okp1_{cmΔ} or Okp1_{nnΔ} is not impaired severely. By introducing the general phenotypes of the Okp1 mutants early in the manuscript, we hope to clarify this point.

2nd Editorial decision - acceptance

08 September 2017

Thank you for submitting your final revised manuscript for our consideration. I am pleased to inform you that we in light of the positive re-assessments (see below) by our three original reviewers, we have now accepted it for publication in The EMBO Journal. Thank you again for this contribution to The EMBO Journal, and congratulations on a successful publication! Please consider us again in the future for your most exciting work.

REFEREE REPORTS

Referee #1:

The authors have satisfactorily addressed my comments made in response to the initial submission. I think the formatting changes improve the readability of the paper, and would now support publication.

Referee #2:

The authors have restructured the manuscript to improve clarity and have included additional

experimental data that support their original conclusions or lead to minor corrections. In my opinion, they have addressed the reviewers' points satisfactory.

Referee #3:

Authors substantially revised their manuscript, combined with additional experiments. Although I gave some minor comments on terminology in introduction, this manuscript should be accepted for publication in EMBO J in principle. My comments are not central to this paper. But as readers are not familiar with yeast names, authors need detail explanations.

1. On page 3, when authors firstly refer COMA, they should describe that COMA is counterpart of the CENP-O complex in vertebrates. While I notice that authors explain this in later part, it may be better to state this explanation, here.

2. On page 3 line 83-85, when authors firstly refer CTF19, they should explain that CTF19 is equivalent to CCAN in human. If they introduced this explanation into line 85 (after ... Schleiffer et al., 2012), next sentence would be "The specific functional relevance of several CCAN subunits is unclear" (not CTF19 subunits).

3. On page 5, when they refer "MIND," they should explain that "MIND" is counterpart of human Mis12 complex."

Corresponding Author Name: Florian Schmitzberger

Journal Submitted to: The EMBO Journal

Manuscript Number: EMBOJ-2017-96636